

# Understanding and utilizing the inner bonds of process tensors

**Moritz Cygorek[1,2] and Erik M. Gauger[2]**

**1** Condensed Matter Theory, Technical University of Dortmund, 44227 Dortmund, Germany
**2** SUPA, Institute of Photonics and Quantum Sciences, Heriot-Watt University,
Edinburgh, EH14 4AS, United Kingdom

## Abstract

Process tensor matrix product operators (PT-MPOs) enable numerically exact simulations for an unprecedentedly broad range of open quantum systems. By representing environment influences in MPO form, they can be efficiently compressed using established algorithms. The dimensions of inner bonds of the compressed PT-MPO may be viewed as an indicator of the complexity of the environment. Here, we show that the inner bonds themselves, not only their dimensions, have a concrete physical meaning: They represent the subspace of the full environment Liouville space which hosts environment excitations that may influence the subsequent open quantum systems dynamics the most. This connection can be expressed in terms of lossy linear transformations, whose pseudoinverses facilitate the extraction of environment observables. We demonstrate this by extracting the environment spin of a central spin problem, the current through a quantum system coupled to two leads, the number of photons emitted from quantum emitters into a structured environment, and the distribution of the total absorbed energy in a driven non-Markovian quantum system into system, environment, and interaction energy terms. Numerical tests further indicate that different PT-MPO algorithms compress environments to similar subspaces. Thus, the physical interpretation of inner bonds of PT-MPOs both provides a conceptional understanding and it enables new practical applications.

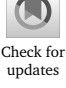

# 1 Introduction

All real-world quantum systems are invariably coupled to their surrounding environment. When the coupling is weak, the Born-Markov approximation provides a sufficient description of the open quantum systems dynamics in the form of Lindblad master equations [1]. However, in many cases, as in charge or excitation transfer in chemical [2] or biological systems [3], in spin systems [4, 5], in solid-state quantum emitters [6–9], or superconducting qubits [10, 11], the system-environment coupling is strong enough to lead to sizable non-Markovian memory effects. Because the Born-Markov approximation becomes insufficient in such situations, predicting the dynamics of open quantum systems then requires methods that accurately account for effects such as renormalization of system energy scales [12, 13], environment-assisted transitions [3, 14, 15], non-exponential decay [16, 17], the formation of quasi-particles like polarons [12], and deviations of multi-time correlation functions from predictions based on the quantum regression theorem [18].

Process tensor matrix product operators (PT-MPOs) provide an attractive and practical solution to tackle such challenging problems. They can be understood as efficient representations of Feynman-Vernon influence functionals [19] [see Fig. 1(a)] in the form of matrix product operators (MPOs) [20, 21] [see Fig. 1(b)] or tensor trains [22]. PT-MPOs can be used to simulate open quantum systems numerically exactly, i.e. they include all effects generated by the microscopic system and environment Hamiltonians to all orders in the coupling. Any inaccuracy is then purely the result of insufficient numerical convergence and can, in principle, be made arbitrarily small by choosing more stringent convergence parameters.

Owing to their practicality and efficiency, PT-MPOs have seen wide adoption within a few years of their inception [24, 25]. For example, Denning *et al.* [9], Fux *et al.* [26], and Vannucci and Gregersen [27] have used PT-MPOs to study the dynamics of semiconductor quantum dots interacting with a phonon bath, while Richter and Hughes have described two emitters coupled to a common waveguide [28]. We recently demonstrated a divide-and-conquer algorithm for constructing periodic PT-MPOs [29], which has enabled million-time-step simulations, e.g., for modelling experiments measuring quantum dot emission spectra after time-dependent (pulsed) driving by Boos *et al.* [30] as well as the analysis of two-color excitation with strongly off-resonant laser pulses by Bracht *et al.* [31]. Link, Tu, and Strunz described a method to calculate periodic PT-MPOs with linear scaling with respect to memory time [32]. Impurity problems have been adressed by Abanin *et al.* [33] and Reichman *et al.* [34] using

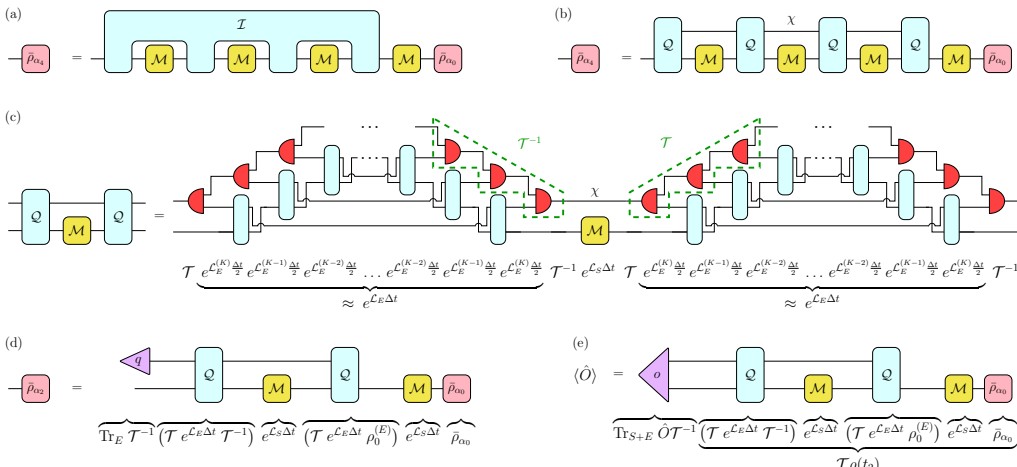

Figure 1: (a) Time-discretized Feynman-Vernon path integral for the reduced system density matrix after $n = 4$ time steps $\bar{\rho}_{\alpha_4}$, which involves the free system propagator $\mathcal{M} = e^{\mathcal{L}_S \Delta t}$ and the influence functional $\mathcal{I}$. The comb-like geometry is required to account for the time-non-local information flow through non-Markovian (finite-memory) environments. (b) The PT-MPO representation of the influence functional with matrices $\mathcal{Q}$ facilitates easy evaluation of the same path integral as a sequence of matrix multiplications. The non-local information flow is now mediated by the inner bonds with maximal dimension $\chi$. (c) Section of a PT-MPO obtained using *Automated Compression of Environments* (ACE) [23]: A PT-MPO for an environment composed of $K$ non-interacting modes is constructed by combining and compressing PT-MPOs of the individual modes one by one. The PT-MPO of a single mode $k$ is generated by the $k$-th environment Liouvillian $e^{\mathcal{L}_E^{(k)} \Delta t}$, where the inner bonds span the $k$-th environment Liouville space. A central finding of this article is that the matrices $\mathcal{Q}$ of the overall PT-MPO can be explicitly linked to the full environment Liouville propagator via $\mathcal{Q} = \mathcal{T} e^{\mathcal{L}_E \Delta t} \mathcal{T}^{-1}$, where $\mathcal{T}$ are lossy (rank-reducing) transformations and $\mathcal{T}^{-1}$ are their pseudoinverses. These are themselves of MPO form. With this insight, the utility of the inner bonds is not limited to truncating PT-MPOs at earlier time steps (d); It also enables the extraction of environment and mixed system-environment observables (e).

fermionic PT-MPOs. Examples involving PT-MPOs for interacting boson and spin environments have been given by Ye and Chan [35] and Guo *et al.* [36], respectively. Moreover, the PT-MPO formalism remains numerically exact when open quantum systems coupled to multiple environments are simulated by interleaving the corresponding PT-MPOs, which are constructed independently of each other. This has been exploited for investigations of non-additive effects of two non-Markovian baths [37] as well as of cooperative effects in multi-quantum-emitter systems [17], and also paves the way for scalable numerically exact simulations of quantum networks [38]. With the algorithm *Automated Compression of Environments* (ACE) [23], we have recently extended the scope of process tensor methods to environments with arbitrary Hamiltonians composed of independent modes. Despite the emerging broad utility and practical power of PT-MPOs, an understanding of their inner structure and information content remains lacking.

Originally, process tensors (PTs) were introduced by Pollock *et al.* in Ref. [24] as a means to characterize non-Markovian quantum processes. As such, they generalize quantum channels, which describe how quantum systems change over a fixed time interval, to a description spanning multiple time steps. To tackle the exponential growth of PTs with the number of time

steps, the PT—a one-dimensional object in time—is mapped to the state of a spatially one-dimensional many-body quantum system via a generalized Choi-Jamiołkowski state-channel isomorphism [39, 40]. This Choi state is then efficiently represented as a matrix product operator (MPO). Contemporarily, Strathearn *et al.* [41] developed TEMPO, which uses MPOs to improve the performance and resolvable memory duration of the QUAPI [42, 43] variant of Feynman-Vernon [19] path integral simulations. The fact that PTs are equivalent to Feynman-Vernon influence functionals for spin-boson-type environments was observed by Jørgensen and Pollock in Ref. [25]. There, it was also found that TEMPO corresponds to a particular contraction of a tensor network that also describes the PT-MPO, and that it is typically more efficient to directly calculate the latter. Thereby, Ref. [25] has laid the grounds for using PT-MPOs as practical tools to simulate open quantum systems, beyond its original purpose for characterizing unknown non-Markovian quantum processes [10, 24].

Invariably, the dimensions $\chi$ of the inner MPO bonds [see Fig. 1(b)] play a crucial role, as they are a key factor for the required computational resources, e.g., for storing the PT-MPO in memory and performing operations with it. Formally, the bond dimensionality can be linked to the Rényi entropy of the Choi state [44], borrowing from one-dimensional many-body MPO theory [45]. Further, recent work provided an analytic bound for the PT-MPO bond dimension for an Ohmic spin-boson environment at zero temperature [46]. There exists broad consensus [24, 29, 44, 46] on the interpretation of the bond dimension $\chi$ as a measure for the simulation complexity of an open quantum system, with some authors also seeking links between $\chi$ and notions of non-Markovianity [47], or of quantum chaos [48].

By constrast, the inner bonds of many-body matrix product states are known to contain more physical information than just their dimensions. Most notably, the entanglement spectrum [49] as well as the transformation behavior of inner bonds of matrix product states can be used to identify topological order [50]. Whether or not the inner bonds of PT-MPOs also contain meaningful physical information beyond their dimensions is not obvious from the derivation of PT-MPO algorithms based on Feynman-Vernon path integral expressions [25, 29, 32]. There, by construction, the inner bonds are used to encode long-range temporal correlations of the environment. This is different in the ACE algorithm [23], where the PT-MPO is constructed directly from the microscopic Hamiltonian of a general environment composed of mutually noninteracting modes. PT-MPOs for individual modes are obtained by exponentiating their respective Liouvillians, before these individual PT-MPOs are combined and compressed to form the PT-MPO for the full environment [see Fig. 1(c)]. Before compression, the inner bonds of the individual mode PT-MPOs simply denote the Liouville space, i.e. the squared Hilbert space, of the respective environment mode, which carries complete information about the state of the latter. It can thus be expected that some of this information survives MPO compression.

In this article, we provide a precise interpretation of the meaning of the inner bonds of PT-MPOs and thereby of the PT-MPOs themselves. By tracking how the inner bonds are transformed in every step of the ACE algorithm [23], we identify the connection between the inner bonds of the final (compressed) PT-MPO and the Liouville space of the environment in terms of lossy linear transformations [see Fig. 1(c)]. Along with the explicit transformation matrices $\mathcal{T}$ (which vary from time step to time step), we obtain corresponding pseudoinverses $\mathcal{T}^{-1}$. Our main result is a conceptually clear interpretation of PT-MPOs: The matrices $\mathcal{Q}$ forming the bulk of the PT-MPO

$$\mathcal{Q} = \mathcal{T} e^{\mathcal{L}_E \Delta t} \mathcal{T}^{-1}, \tag{1}$$

are simply the propagators $e^{\mathcal{L}_E \Delta t}$ of the full environment over a time step $\Delta t$ compressed to the relevant (contemporary) subspace via $\mathcal{T}$ and $\mathcal{T}^{-1}$, where $\mathcal{L}_E$ is the environment Liouvillian in the superoperator notation described in section 2.1. This relevant subspace is implicitly and automatically determined by MPO compression.

As a consequence, the inner bonds carry time-local information about the dynamical evolution of the environment, which can be partially reconstructed with the help of the pseudoinverses $\mathcal{T}^{-1}$ as indicated in Fig. 1(e). The extraction of environment and mixed system-environment observables $\mathcal{O}$ is achieved by terminating the PT-MPO path sum with an object $o$, which we call an *observable closure* and which expresses the action of $o \cdot = \text{Tr}_{S+E}\{\mathcal{O}\mathcal{T}^{-1}\cdot\}$, where $\text{Tr}_{S+E}$ denotes the trace over system and environment degrees of freedom. In order to avoid the explicit storage of the matrices $\mathcal{T}^{-1}$ at each time step, we instead transform $o$ along the individual steps of the PT-MPO construction in the ACE algorithm.

However, because the transformation $\mathcal{T}$ is lossy, there is no guarantee that a given environment observable can indeed be accurately reconstructed. Comparing convergence of different environment observables therefore allows one to probe what information is conveyed in the inner bonds of PT-MPOs and what is eliminated by MPO compression. The comparison with a hierarchy of Heisenberg equations of motion for operator averages reveals that first order system-environment correlations that directly enter the Heisenberg equation of motion for system observables are much more accurately reproduced than environment observables further down the hierarchy that affect the system only indirectly via influencing the first order correlations. We shall presently demonstrate that this insight can be utilized to devise alternative, more accurate schemes to extract environment observables from PT-MPO simulation.

We consider a variety of examples showcasing available access to environmental and mixed observables. First, on the example of a central spin model with total spin conservation, we show that the total environment spin can be faithfully obtained from the inner bonds of PT-MPOs. Next, we demonstrate the extraction of currents through a central site coupled to two metallic leads at different chemical potentials. The convergence of environment observables is then analyzed in more detail on the example of photon emission from a quantum emitter. Finally, we calculate the total energy absorbed by an externally driven non-Markovian open quantum system as well as its distribution into terms associated to only the system, to only the environment, as well as to the system-environment interaction term, which is proportional to system-environment correlations. On the last example, we also perform a numerical experiment where we take the observable closures obtained from ACE simulations and, after fixing the gauge freedom, apply them to inner bonds of PT-MPOs calculated using the algorithm by Jørgensen and Pollock [25], which is based on the Feynman-Vernon path integral expression for the influence functional and never makes reference to any particular environment Liouville space. Yet, we find a remarkable agreement between the extracted environment observables from both PT-MPOs, indicating that the space described by the inner bonds is essentially independent of the PT-MPO algorithm, and hence a universal property of PT-MPOs.

This article is structured as follows: First, in Section 2, we summarize and rederive the ACE algorithm using a superoperator notation, which differs from the original derivation in Ref. [23] to clearly reveal the connection between the full environment Liouville space and the inner bonds of ACE PT-MPOs before MPO compression. In Section 3, we explicitly derive the transformation matrices corresponding to the compression of inner indices during the ACE algorithm. We then discuss the resulting overall transformation matrices as well as their pseudoinverses in Section 3.3, before describing how their explicit storage can be avoided by tracking the transformation of observable closures in Section 3.4. The above-mentioned series of examples is presented in Section 4. Finally, our findings are summarized and discussed in Section 5.

## 2 Theoretical background

We first introduce the Liouville space notation used throughout the paper and summarize the basic concepts of influence functionals, PT-MPOs, and the ACE algorithm.

### 2.1 Notation

The time evolution of a density matrix of a general closed quantum system can be conveniently expressed in the superoperator formalism. To this end, we utilize the isomorphism between the Hilbert space $\mathcal{H}$, which hosts "ket" states $|\nu\rangle$, and its dual $\mathcal{H}^*$, which contains "bra" states $\langle\nu|$. This enables the mapping of density matrices in the space $\mathcal{H}\otimes\mathcal{H}^*$ onto vectors in the squared Hilbert space $\mathcal{H}\otimes\mathcal{H}$, which we refer to as the Liouville space, by

$$\hat{\rho} = \sum_{\nu,\mu=0}^{\dim(\mathcal{H})-1} \rho_{\nu\mu}|\nu\rangle\langle\mu| \qquad \longleftrightarrow \qquad |\rho) = \sum_{\nu,\mu=0}^{\dim(\mathcal{H})-1} \rho_{\nu\mu}|\nu\rangle\otimes|\mu\rangle = \sum_{\alpha=0}^{(\dim(\mathcal{H}))^2-1} \rho_\alpha|\alpha), \quad (2)$$

where we have introduced combined indices $\alpha = (\nu,\mu)$ and defined the set of basis vectors $|\alpha) = |\nu\rangle\otimes|\mu\rangle$ of the Liouville space $\mathcal{H}\otimes\mathcal{H}$. Thereby, the Liouville-von Neumann equation takes the form of a matrix-vector product

$$\frac{\partial}{\partial t}\hat{\rho} = -\frac{i}{\hbar}\big[H,\hat{\rho}\big] = -\frac{i}{\hbar}\big(H\hat{\rho}-\hat{\rho}H\big) \qquad \longleftrightarrow \qquad \frac{\partial}{\partial t}|\rho) = -\frac{i}{\hbar}\big(H\otimes\mathbb{1}-\mathbb{1}\otimes H^T\big)|\rho) = \mathcal{L}|\rho), \quad (3)$$

with formal solution

$$|\rho(t)) = e^{\mathcal{L}t}|\rho(0)). \quad (4)$$

The inclusion of additional Markovian loss or decoherence processes via Lindblad terms and extension to a time-dependent Hamiltonian is straightforward.

We now consider a general open quantum system, where the composite system $\mathcal{H} = \mathcal{H}_S\otimes\mathcal{H}_E$ can be decomposed into system $\mathcal{H}_S$ and environment subspaces $\mathcal{H}_E$, respectively. Henceforth, system Hilbert space basis states will be denoted by $|\nu\rangle$ and $|\mu\rangle$, while $|\xi\rangle$ and $|\eta\rangle$ will refer to basis states of the environment Hilbert space. Instead of applying the superoperator mapping in Eq. (2) directly to the total Hilbert space $\mathcal{H}$, we choose to first rearrange indices in such a way that system $\alpha = (\nu,\mu)$ and environment degrees of freedom $\beta = (\xi,\eta)$ remain separate from each other:

$$\hat{\rho} = \sum_{\nu,\mu}\sum_{\xi,\eta}\rho_{\nu,\xi,\mu,\eta}(|\nu\rangle\otimes|\xi\rangle)(\langle\mu|\otimes\langle\eta|) \qquad \longleftrightarrow \qquad |\rho) = \sum_{\nu,\mu}\sum_{\xi,\eta}\rho_{(\nu,\mu),(\xi,\eta)}|\nu\rangle\otimes|\mu\rangle\otimes|\xi\rangle\otimes|\eta\rangle$$
$$= \sum_{\alpha,\beta}\rho_{\alpha,\beta}|\alpha)\otimes|\beta)$$
$$= \sum_{\alpha,\beta}\rho_{\alpha,\beta}|\alpha,\beta). \quad (5)$$

In the final line of the above equation we have defined the components of the total density matrix $\rho_{\alpha,\beta} = (\alpha,\beta|\rho)$, where $(\alpha,\beta| = (\alpha|\otimes(\beta|$ with $(\alpha| = \langle\nu|\otimes\langle\mu|$ and $(\beta| = \langle\xi|\otimes|\langle\eta|$. The time-evolution of the total density matrix is then formally given by integrating the Liouville-von Neumann equation

$$\rho_{\alpha,\beta}(t) = \sum_{\alpha_0,\beta_0}(\alpha,\beta|e^{\mathcal{L}t}|\alpha_0,\beta_0)\rho_{\alpha_0,\beta_0}(0), \quad (6)$$

where $\rho_{\alpha_0,\beta_0}(0)$ are the coefficients of the joint initial state. To extract a physical observable characterized by the operator $\hat{O}$ acting on the total Hilbert space $\mathcal{H}$, we define

$$(\hat{O}| = \sum_{\alpha,\beta} o_{\alpha,\beta}(\alpha,\beta|, \tag{7}$$

$$o_{(\nu,\mu),(\xi,\eta)} = \left(\langle\mu|\otimes\langle\eta|\right)\hat{O}\left(|\nu\rangle\otimes|\xi\rangle\right), \tag{8}$$

so that

$$(\hat{O}|\rho) = \sum_{\alpha,\beta} o_{\alpha,\beta}\rho_{\alpha,\beta} = \mathrm{Tr}\left\{\hat{O}\hat{\rho}\right\} = \langle\hat{O}\rangle. \tag{9}$$

Specifically, system observables $\hat{O}_S$ acting only on $\mathcal{H}_S$ are obtained by

$$\langle\hat{O}_S\rangle = (\hat{O}_S\otimes\mathbb{1}_E|\rho) = \sum_{\alpha,\beta} o_\alpha\mathfrak{I}_\beta\rho_{\alpha,\beta} = \sum_\alpha o_\alpha\bar{\rho}_\alpha, \tag{10}$$

where $\bar{\rho}_\alpha = \sum_\beta \mathfrak{I}_\beta\rho_{\alpha,\beta}$ is the reduced system density matrix, $o_\alpha = o_{(\nu,\mu)} = \langle\mu|\hat{O}_S|\nu\rangle$, and $\mathfrak{I}_\beta = \mathfrak{I}_{(\xi,\eta)} = \delta_{\xi,\eta}$ denotes the trace operation on the environment subspace. Analogously, we define $\mathfrak{I}_\alpha = \mathfrak{I}_{(\nu,\mu)} = \delta_{\nu,\mu}$ to describe the trace over the system degrees of freedom.

## 2.2 Influence functionals

Our goal is to integrate the Liouville-von Neumann equation (6) without explicitly operating on the full environment Liouville space because this is generally numerically infeasible. Instead, an exact representation of the effects of the environment can be formulated, where the explicit environment degrees of freedom are traced out. The corresponding object is the *influence functional*, which was first derived by Feynman and Vernon in the context of real-time path integrals [19].

However, we shall now instead consider an alternative derivation within the superoperator formalism that is obtained in three steps: First, time is discretized on a regular grid $t_j = t_0 + j\Delta t$ with time steps $\Delta t$, yielding the time evolution

$$(\alpha_n,\beta_n|e^{\mathcal{L}t_n}|\alpha_0,\beta_0) = \sum_{\substack{\alpha_{n-1},\dots,\alpha_1 \\ \beta_{n-1},\dots,\beta_1}} \prod_{l=1}^{n}(\alpha_l,\beta_l|e^{\mathcal{L}\Delta t}|\alpha_{l-1},\beta_{l-1}). \tag{11}$$

To simplify notation, time arguments are henceforth implied in the index labels. For example, the subscript $j$ on the system Liouville space index $\alpha_j$ indicates that $\bar{\rho}_{\alpha_j}$ denotes $\bar{\rho}_{\alpha_j}(t_j)$, i.e. the reduced system density matrix at time $t_j$.

Second, the total Liouvillian is decomposed into $\mathcal{L} = \mathcal{L}_S + \mathcal{L}_E$, where the system Liouvillian $\mathcal{L}_S$ only affects the system, while the environment Liouvillian $\mathcal{L}_E$ includes the system-environment interaction and, hence, affects both system and environment. The time evolution within each time step is then split using the Trotter decomposition

$$e^{(\mathcal{L}_E+\mathcal{L}_S)\Delta t} = e^{\mathcal{L}_E\Delta t}e^{\mathcal{L}_S\Delta t} + \mathcal{O}(\Delta t^2). \tag{12}$$

The implementation of a symmetric Trotter decomposition with error $\mathcal{O}(\Delta t^3)$ is straightforward, but we proceed our derivation with Eq. (12) for a shorter notation.

Finally, assuming that the initial state $\rho_{\alpha_0,\beta_0} = \bar{\rho}_{\alpha_0}\rho_{\beta_0}^E$ factorizes into system $\bar{\rho}_{\alpha_0}$ and environment parts $\rho_{\beta_0}^E$, one traces over the environment at the final time step. Then, the reduced system density matrix at time $t = t_n$ can be expressed as

$$\bar{\rho}_{\alpha_n} = \sum_{\substack{\alpha_{n-1},\dots,\alpha_0 \\ \alpha'_n,\dots,\alpha'_1}} \mathcal{I}^{(\alpha_n,\alpha'_n)\dots(\alpha_1,\alpha'_1)}\left(\prod_{l=1}^{n}\mathcal{M}^{\alpha'_l\alpha_{l-1}}\right)\bar{\rho}_{\alpha_0}, \tag{13}$$

where $\mathcal{M}^{\alpha'_l \alpha_{l-1}} = (\alpha'_l | e^{\mathcal{L}_S \Delta t} | \alpha_{l-1})$ denotes the free system propagator and

$$\mathcal{I}^{(\alpha_n, \alpha'_n)\dots(\alpha_1, \alpha'_1)} = \sum_{\beta_n,\dots,\beta_1} \mathfrak{I}_{\beta_n} \left( \prod_{l=1}^{n} (\alpha_l, \beta_l | e^{\mathcal{L}_E \Delta t} | \alpha'_l, \beta_{l-1}) \right) \rho^E_{\beta_0}, \tag{14}$$

is the influence functional. The sum over all possible combinations of system indices $\alpha_l$ and $\alpha'_l$ in Eq. (13) becomes the integral over all system paths in the continuous-time limit described by the Feynman-Vernon real-time path integral formalism [19].

If the environment is Gaussian, as in the case of the spin-boson model, explicit expressions for the influence functional can be obtained by analytically integrating over the environment degrees of freedom $\beta_l$ [19], which is used in various algorithms including QUAPI [42], TEMPO [41], and others [25,29,32]. A numerical approach for more general environments is provided by ACE [23], where environment influences are calculated explicitly using Eq. (14). A key element to make ACE numerically feasible is the MPO representation of the environment influences, which we discuss next.

## 2.3 Process tensor matrix product operators

The Feynman-Vernon path sum in Eq. (13) yields an exact description of the open quantum systems dynamics in the limit $\Delta t \to 0$. However, its practical application is limited by the exponential scaling of the number of paths $\mathcal{O}(\dim(\mathcal{H}_S)^{4n})$ with the number of time steps $n$. A practical solution is provided by PT-MPOs [24,25], which represent influence functionals efficiently in the form of MPOs [20,21]

$$\mathcal{I}^{(\alpha_n, \alpha'_n)\dots(\alpha_1, \alpha'_1)} = \sum_{d_n,\dots,d_0} \mathcal{Q}^{(\alpha_n, \alpha'_n)}_{d_n d_{n-1}} \mathcal{Q}^{(\alpha_{n-1}, \alpha'_{n-1})}_{d_{n-1} d_{n-2}} \cdots \mathcal{Q}^{(\alpha_1, \alpha'_1)}_{d_1 d_0}. \tag{15}$$

Here, $\mathcal{Q}^{(\alpha_l, \alpha'_l)}_{d_l d_{l-1}}$ are interpreted as matrices with respect to inner bond indices $d_l$, where bond indices at the edges take only one value $d_0 = d_n = 0$. The MPO form makes it possible to perform the path summation in Eq. (13) step by step. To this end, we define the extended density matrix $\rho^\alpha_d$ by the iteration

$$\rho^{\alpha_0}_0 = \bar{\rho}_{\alpha_0}, \tag{16a}$$

$$\rho^{\alpha_l}_{d_l} = \sum_{\alpha'_l, \alpha_{l-1}, d_{l-1}} \mathcal{Q}^{(\alpha_l, \alpha'_l)}_{d_l d_{l-1}} \mathcal{M}^{\alpha'_l \alpha_{l-1}} \rho^{\alpha_{l-1}}_{d_{l-1}}, \tag{16b}$$

which turns the sum over exponentially many paths into a linear number of matrix multiplications. The reduced system density matrix at the last time step $t_n$ is then given by $\bar{\rho}_{\alpha_n} = \rho^{\alpha_n}_0$. At intermediate time steps it can be obtained by $\bar{\rho}_{\alpha_l} = \sum_{d_l} q_{d_l} \rho^{\alpha_l}_{d_l}$, where the closures $q_{d_l}$ are constructed from the PT-MPO as described in Ref. [23]. Thus, the brunt of the work required to simulate the open quantum system is now shifted to bringing the influence functional into the PT-MPO form of Eq. (15).

In principle, any tensor can be brought into MPO form by successive Schmidt decompositions or, equivalently, singular value decompositions (SVDs). It is straightforward to show [20] that this provides an upper bound for the inner bond dimensions by $d_j, d_{n-j} \leq \dim(\mathcal{H}_S^{4j})$, which is maximal at the center of the chain, e.g., $d_{n/2} \leq \dim(\mathcal{H}_S^{2n})$ for even $n$. However, often, many of the singular values are zero or negligibly small, which reduces the inner bond dimensions significantly. Furthermore, to avoid exponential scaling incurred by the factorization of the full tensor, PT-MPOs are usually built up from smaller blocks, keeping them in MPO form at all times [23,25,29]. After adding new blocks, the PT-MPO is compressed by sweeping along the MPO chain and reducing inner dimensions using truncated SVDs, where singular values below

a given threshold are neglected [21]. For example, converged results have been obtained in typical applications with compressed PT-MPOs with maximal inner dimensions $\chi$ in the range of a few dozen to several hundreds [23, 29].

## 2.4 Automated compression of environments

The ACE algorithm [23] enables the calculation of PT-MPOs also for non-Gaussian environments, for which no analytical expressions based on path integration are available. Instead, Eq. (14) is directly brought into the MPO form of Eq. (15) starting from the microscopic Hamiltonian. ACE is based on the observation that, in our Liouville space notation, an exact PT-MPO can be formally constructed by setting

$$\mathcal{Q}^{(\alpha_l,\alpha_l')}_{\beta_l,\beta_{l-1}} = (\alpha_l,\beta_l|e^{\mathcal{L}_E\Delta t}|\alpha_l',\beta_{l-1}),$$
(17a)

for matrices inside the MPO chain $2 \leq l \leq n-1$ and

$$\mathcal{Q}^{(\alpha_1,\alpha_1')}_{\beta_1,\beta_0} = \sum_{\beta'}(\alpha_1,\beta_1|e^{\mathcal{L}_E\Delta t}|\alpha_1',\beta')\rho^E_{\beta'}\,\delta_{\beta_0,0}\,,$$
(17b)

and

$$\mathcal{Q}^{(\alpha_n,\alpha_n')}_{\beta_n,\beta_{n-1}} = \delta_{\beta_n,0}\sum_{\beta'}\mathfrak{I}_{\beta'}(\alpha_n,\beta'|e^{\mathcal{L}_E\Delta t}|\alpha_n',\beta_{n-1})\,,$$
(17c)

for the first and the last MPO matrix, respectively.

Note, however, that the inner bond dimension of this PT-MPO is equal to the dimension of the full environment Liouville space $\dim(\mathcal{H}_E)^2$. Hence, the propagatation using the iteration Eq. (16) is as computationally expensive as solving the time evolution of the total system comprised of system of interest and environment in Liouville space.

ACE overcomes this challenge by considering the environment as being composed of $N_E$ independent degrees of freedom, henceforth referred to as *modes*. The total environmental Hilbert space is then a product of individual subspaces $\mathcal{H}_E = \mathcal{H}_E^{(1)} \otimes \mathcal{H}_E^{(2)} \otimes \ldots \mathcal{H}_E^{(N_E)}$, whilst the environment Hamiltonian is given by the sum $H_E = \sum_{k=1}^{N_E} H_E^{(k)}$, where each summand $H_E^{(k)}$ only operates on the system and the $k$-th environment mode Hilbert space $\mathcal{H}_S \otimes \mathcal{H}_E^{(k)}$. Consequently, the total Liouvillian can also be written as a sum of $N_E$ terms $\mathcal{L}_E = \sum_k^{N_E} \mathcal{L}_E^{(k)}$, where the $k$-th environment Liouvillian has the explicit matrix representation

$$(\alpha,\beta^{(k)}|\mathcal{L}_E^{(k)}|\alpha',\beta'^{(k)}) = ((\nu,\mu),(\xi^{(k)},\eta^{(k)})|\mathcal{L}_E^{(k)}|(\nu',\mu'),(\xi'^{(k)},\eta'^{(k)}))$$
$$= -\frac{i}{\hbar}\Big[\langle\nu,\xi^{(k)}|H_E^{(k)}|\nu',\xi'^{(k)}\rangle\delta_{\mu,\mu'}\delta_{\eta^{(k)},\eta'^{(k)}}$$
$$- \langle\mu',\eta'^{(k)}|H_E^{(k)}|\mu,\eta^{(k)}\rangle\delta_{\nu,\nu'}\delta_{\xi^{(k)},\xi'^{(k)}}\Big]\,,$$
(18)

where indices $\beta^{(k)} = (\xi^{(k)},\eta^{(k)})$ enumerate a basis of states for the $k$-th environment Liouville space. The $k$-th environment propagator $(\alpha,\beta^{(k)}|e^{\mathcal{L}_E^{(k)}\Delta t}|\alpha',\beta'^{(k)})$ is then given by the matrix exponential of Eq. (18).

For each individual environment mode $k$, a PT-MPO is constructed using Eqs. (17) with the Liouvillian $\mathcal{L}_E$ replaced by $\mathcal{L}_E^{(k)}$. It is assumed that the inner bond dimension, which now corresponds to the Liouville space of only a single environment mode, is manageable. The PT for the full environment is obtained by combining the PTs for all individual modes based on the sequential application of the symmetric Trotter decomposition,

$$e^{\sum_{k=1}^{K}\mathcal{L}_E^{(k)}\Delta t} \approx e^{\mathcal{L}_E^{(K)}\frac{\Delta t}{2}}\left(e^{\sum_{k=1}^{K-1}\mathcal{L}_E^{(k)}\Delta t}\right)e^{\mathcal{L}_E^{(K)}\frac{\Delta t}{2}}\,,$$
(19)

for $K = 2, \ldots N_E$, which incurs a Trotter error $\mathcal{O}(\Delta t^3)$.

To keep the bond dimensions reasonably small, the PT-MPO is compressed after the inclusion of each additional environment mode, as described in detail in the next section. Eventually, after having iteratively incorporated the PT-MPO of each environmental mode, one arrives at a PT-MPO describing all effects of the microscopic Hamiltonian numerically exactly, where the only numerical errors are due to the Trotter decomposition and the MPO compression.

# 3 Transformation of inner bonds of process tensors

Our goal now is to explicitly link the inner bonds of PT-MPOs $\mathcal{Q}_{d_l,d_{l-1}}^{(\alpha_l,\alpha_l')}$ to the environment propagator in Liouville space. Concretely, we show that the PT-MPO matrices obtained using ACE can be written as

$$\mathcal{Q}_{d_l,d_{l-1}}^{(\alpha_l,\alpha_l')} = \sum_{\beta_l,\beta_{l-1}} \mathcal{T}_{d_l,\beta_l}(\alpha_l,\beta_l|e^{\mathcal{L}_E \Delta t}|\alpha_l',\beta_{l-1})\mathcal{T}_{\beta_{l-1},d_{l-1}}^{-1}, \tag{20}$$

where $\mathcal{T}_{d_l,\beta_l}$ describes a lossy transformation and $\mathcal{T}_{\beta_{l-1},d_{l-1}}^{-1}$ is a pseudoinverse of the corresponding transformation matrix $\mathcal{T}_{d_{l-1},\beta_{l-1}}$ at the previous time step $t_{l-1}$. To this end, we follow the evolution of the transformation matices $\mathcal{T}_{d_l,\beta_l}$ and their pseudoinverses in every step of the ACE algorithm.

As a corollary of Eq. (20), we observe that the propagated quantity $\rho_{d_l}^{\alpha_l}$ in Eq. (16) corresponds to the total density matrix after compression of the inner bonds $\rho_{d_l}^{\alpha_l} = \sum_{\beta_l} \mathcal{T}_{d_l,\beta_l}\rho_{\alpha_l,\beta_l}$, which can be written in the notation of section 2.1 as $|\mathcal{T}\rho(t))$. This suggests that general observables $\hat{O}$, including environment and mixed system-environment observables, can be inferred from

$$\langle \hat{O}(t) \rangle \approx (\hat{O}\mathcal{T}^{-1}|\mathcal{T}\rho(t)), \tag{21}$$

which, however, only holds approximately due to the lossy nature of the transformation $\mathcal{T}$. Its validity is tested on several examples in Section 4.

The advantage of Eq. (21) is that it provides an efficient way to extract environment observables without the need store the lossy transformation matrices $\mathcal{T}$ and $\mathcal{T}^{-1}$ explicitly. It suffices to track only the behaviour of $(\hat{O}\mathcal{T}^{-1}|$ for given observables $\hat{O}$.

## 3.1 PT-MPO combination

We now describe how the transformations $\mathcal{T}$ and $\mathcal{T}^{-1}$ in Eq. (20) are affected in the individual combination and compression steps of the ACE algorithm [23]. We do this by induction: We start with the trivial PT-MPO with matrices $\mathcal{Q}_{d_l,d_{l-1}}^{(\alpha_l,\alpha_l')} = \delta_{\alpha_l,\alpha_l'}\delta_{d_l,0}\delta_{d_{l-1},0}$, which correspond to a dummy environment mode of dimension $\dim(\mathcal{H}_E^{(0)})=1$ with Hamiltonian $H_E^{(0)} = 0$. The initial transformation matrices are $\mathcal{T}_{d_l,\beta_l} = \mathcal{T}_{\beta_l,d_l}^{-1} = \delta_{d_l,0}\delta_{\beta_l,0}$.

In the induction step, it has to be shown that, if the MPO matrices of the PT accounting for modes $k = 1, \ldots, K-1$ are of the form in Eq. (20), then the resulting MPO matrices of the PT including the influence of the mode $K$ are also of the form in Eq. (20), and the respective transformation matrices are related by well-defined linear operations.

Denoting MPO matrices of the input PT-MPO by $\mathcal{Q}_{d_l,d_{l-1}}^{(\alpha_l,\alpha_l')}$ and that of the resulting PT-MPO by $\tilde{\mathcal{Q}}_{\tilde{d}_l,\tilde{d}_{l-1}}^{(\tilde{\alpha}_l,\tilde{\alpha}_l')}$, it follows from the symmetric Trotter formula in Eq. (19) that

$$\tilde{\mathcal{Q}}_{(d_l,\beta_l),(d_{l-1},\beta_{l-1})}^{(\tilde{\alpha}_l,\tilde{\alpha}_l')} \approx \sum_{\alpha_l,\alpha_l',\beta'} \mathcal{B}_{\beta_l,\beta'}^{(\tilde{\alpha}_l,\alpha_l)} \mathcal{Q}_{d_l,d_{l-1}}^{(\alpha_l,\alpha_l')} \mathcal{B}_{\beta',\beta_{l-1}}^{(\alpha_l',\tilde{\alpha}_l')}, \tag{22}$$

with matrix representation of the $K$-th environment Liouville propagator for half a time step $\mathcal{B}^{(\tilde{\alpha}_l,\alpha_l)}_{\beta_l,\beta'} = (\tilde{\alpha}_l,\beta_l|e^{\mathcal{L}^K_E \frac{\Delta t}{2}}|\alpha_l,\beta')$. The relation to Eq. (20) is established by identifying

$$
\mathcal{Q}^{(\alpha_l,\alpha'_l)}_{d_l,d_{l-1}} \approx \sum_{\substack{\beta^{(1)}_l,\ldots,\beta^{(K-1)}_l \\ \beta^{(1)}_{l-1},\ldots,\beta^{(K-1)}_{l-1}}} \mathcal{T}_{d_l,(\beta^{(1)}_l,\ldots,\beta^{(K-1)}_l)}(\alpha_l,\beta^{(1)}_l,\ldots,\beta^{(K-1)}_l|e^{\sum\limits_{k=1}^{K-1}\mathcal{L}^{(k)}_E \Delta t}|\alpha'_l,\beta^{(1)}_{l-1},\ldots,\beta^{(K-1)}_{l-1})\mathcal{T}^{-1}_{(\beta^{(1)}_{l-1},\ldots,\beta^{(K-1)}_{l-1}),d_{l-1}}
$$

$$
\Longrightarrow \tilde{\mathcal{Q}}^{(\tilde{\alpha}_l,\tilde{\alpha}'_l)}_{\tilde{d}_l,\tilde{d}_{l-1}} \approx \sum_{\substack{\beta^{(1)}_l,\ldots,\beta^{(K)}_l \\ \beta^{(1)}_{l-1},\ldots,\beta^{(K)}_{l-1}}} \tilde{\mathcal{T}}_{\tilde{d}_l,(\beta^{(1)}_l,\ldots,\beta^{(K)}_l)}(\tilde{\alpha}_l,\beta^{(1)}_l,\ldots,\beta^{(K)}_l|e^{\sum\limits_{k=1}^{K}\mathcal{L}^{(k)}_E \Delta t}|\tilde{\alpha}'_l,\beta^{(1)}_{l-1},\ldots,\beta^{(K)}_{l-1})\tilde{\mathcal{T}}^{-1}_{(\beta^{(1)}_{l-1},\ldots,\beta^{(K)}_{l-1}),\tilde{d}_{l-1}},
$$

$$
\tag{23}
$$

where the original transformation matrices $\mathcal{T}$ and $\mathcal{T}^{-1}$ are changed into

$$
\tilde{\mathcal{T}}_{\tilde{d}_l,(\beta^{(1)}_l,\ldots,\beta^{(K)}_l)} = \delta_{\tilde{d}_l,(d_l,\beta^{(K)}_l)}\mathcal{T}_{d_l,(\beta^{(1)}_l,\ldots,\beta^{(K-1)}_l)}, \tag{24a}
$$

$$
\tilde{\mathcal{T}}^{-1}_{(\beta^{(1)}_l,\ldots,\beta^{(K)}_l),\tilde{d}_l} = \mathcal{T}^{-1}_{(\beta^{(1)}_l,\ldots,\beta^{(K-1)}_l),d_l}\delta_{(d_l,\beta^{(K)}_l),\tilde{d}_l}, \tag{24b}
$$

such that the original inner bonds $d_l$ are simply expanded to include the Liouville space of the $K$-th environment mode described by basis states with indices $\beta^{(K)}_l$.

## 3.2 MPO compression

Combining PT-MPOs of individual environment modes increases the inner dimension of the resulting PT-MPO. To keep the combined PT-MPO tractable, it undergoes a compression step after every combination. Compression is achieved by sweeping along the MPO chain, first forward, then backward, while reducing the inner dimension by truncated SVDs [23, 25].

SVDs decompose arbitrary $n \times m$ matrices $A$

$$
A = U\Sigma V^\dagger, \tag{25}
$$

where $U$ and $V$ are matrices with orthogonal columns, while $\Sigma$ is a diagonal matrix with the non-negative real singular values $\sigma_j$ on the diagonal, which are assumed to be sorted in descending order, i.e. $\sigma_j \geq \sigma_k$ for $j \leq k$. Here, we use truncated SVDs by keeping only the $s$ most significant singular values $\sigma_j \geq \epsilon\sigma_0$, where $\epsilon$ is a given truncation threshold relative to the largest singular value $\sigma_0$, and also restricting the matrices $U$ and $V$ to their first $s$ columns. For our applications, $s$ will be the inner bond dimension after PT-MPO compression.

In fact, the Eckart-Young-Mirsky theorem [51] states that SVDs provide the best low-rank approximation to a general matrix for given rank $s$. However, MPO compression at a given site using truncated SVD is only locally optimal if the MPO is in mixed canonical form [21]. One way to achieve this is to first perform a forward sweep with non-truncating SVDs and only truncating during the backward sweep. Yet local optimal compression is not necessarily needed for the most efficient algorithm. Here, we use a strategy in the spirit of the zip-up algorithm of Ref. [52], where truncation is performed for both, forward and backward sweeps. The sizable speed-up achieved by operating on matrices with smaller dimensions is typically well worth the slight increase in numerical error, especially because the latter can be mitigated by using a smaller compression threshold. As discussed in Appendix A of Ref. [29] as well as in Ref. [53], choosing different compression parameters for different sweeps can be used to fine-tune PT-MPO algorithms. For example, using a different threshold $\epsilon_{fw}$ for forward sweeps compared to backward sweeps $\epsilon_{bw}$, one can interpolate between the locally optimal compression ($\epsilon_{fw} = 0$) and the zip-up-like approach ($\epsilon_{fw} = \epsilon_{bw}$). Here, we use $\epsilon_{fw} = \epsilon_{bw} = \epsilon$ throughout this article.

Furthermore, note that MPO compression by SVDs in general is aimed at minimizing the norm distance between MPOs before and after compression. To our knowledge, no formal theory linking this norm distance to a precise error bound for physical observables is available so far. However, numerical simulations in Ref. [53] indicate that the PT-MPO norm distance and system observables show similar convergence behavior. This provides empirical justification for PT-MPO compression using sweeps with SVDs as described above.

Next, we summarize how MPO matrices and in particular their inner bonds are affected by forward and backward sweeps. To simplify the notation, we henceforth imply truncation whenever we refer to SVDs like in Eq. (25).

### 3.2.1 Forward sweep

First, we compress the PT-MPO by sweeping along the MPO in the forward direction, i.e., from $l = 1$ to $l = n$. At step $l$, we perform an SVD on the PT-MPO matrices $\mathcal{Q}^{(\alpha_l, \alpha'_l)}_{d_l, d_{l-1}}$, which we interpret as a single matrix $A_{d_l, ((\alpha_l, \alpha'_l), d_{l-1})}$ with outer indices combined with the right inner index $d_{l-1}$.

$$\mathcal{Q}^{(\alpha_l, \alpha'_l)}_{d_l, d_{l-1}} = \sum_{j_l} \overleftarrow{U}_{d_l j_l} \overleftarrow{\sigma}_{j_l} \overleftarrow{V}^{\dagger}_{j_l ((\alpha_l, \alpha'_l), d_{l-1})} . \tag{26}$$

After truncation, the MPO matrix at steps $l$ is replaced by

$$\tilde{\mathcal{Q}}^{(\alpha_l, \alpha'_l)}_{j_l, d_{l-1}} = \overleftarrow{V}^{\dagger}_{j_l ((\alpha_l, \alpha'_l), d_{l-1})} , \tag{27a}$$

while $\overleftarrow{U}_{d_l j_l}$ and the singular values $\overleftarrow{\sigma}_{j_l}$ are passed on to the next MPO matrix at step $l+1$

$$\tilde{\mathcal{Q}}^{(\alpha_{l+1}, \alpha'_{l+1})}_{d_{l+1}, j_l} = \sum_{d_l} \mathcal{Q}^{(\alpha_{l+1}, \alpha'_{l+1})}_{d_{l+1}, d_l} \overleftarrow{U}_{d_l j_l} \overleftarrow{\sigma}_{j_l} . \tag{27b}$$

Passing on the singular values $\overleftarrow{\sigma}_{j_l}$, which are indicators of the local importance of the respective indices $j_l$, is crucial for efficient compression aimed at selecting degrees of freedom that are relevant for describing the system dynamics over many time steps.

Alternatively, the updated MPO matrices can be expressed as

$$\tilde{\mathcal{Q}}^{(\alpha_l, \alpha'_l)}_{j_l, d_{l-1}} = \sum_{d_l} \overleftarrow{T}_{j_l, d_l} \mathcal{Q}^{(\alpha_l, \alpha'_l)}_{d_l, d_{l-1}} , \tag{28a}$$

$$\tilde{\mathcal{Q}}^{(\alpha_{l+1}, \alpha'_{l+1})}_{d_{l+1}, j_l} = \sum_{d_l} \mathcal{Q}^{(\alpha_{l+1}, \alpha'_{l+1})}_{d_{l+1}, d_l} \overleftarrow{T}^{-1}_{d_l, j_l} , \tag{28b}$$

with transformation matrices

$$\overleftarrow{T}_{j_l, d_l} = \overleftarrow{\sigma}^{-1}_{j_l} \overleftarrow{U}^{\dagger}_{j_l d_l} , \tag{29a}$$

$$\overleftarrow{T}^{-1}_{d_l, j_l} = \overleftarrow{U}_{d_l j_l} \overleftarrow{\sigma}_{j_l} . \tag{29b}$$

Assuming that the original PT-MPO matrices have been expressible in terms of environment propagators after lossy compression as in Eq. (20), the corresponding transformation matrices $\mathcal{T}$ and $\mathcal{T}^{-1}$ are themselves transformed as

$$\tilde{\mathcal{T}}_{j_l, \beta_l} = \sum_{d_l} \overleftarrow{T}_{j_l, d_l} \mathcal{T}_{d_l, \beta_l} , \tag{30a}$$

$$\tilde{\mathcal{T}}^{-1}_{\beta_l, j_l} = \sum_{d_l} \mathcal{T}^{-1}_{\beta_l, d_l} \overleftarrow{T}^{-1}_{d_l, j_l} . \tag{30b}$$

### 3.2.2 Backward sweep

Compression is enhanced, if the forward sweep over the PT-MPO is followed by a backward sweep from $l = n$ to $l = 2$. Here, the outer indices of the MPO matrices $\mathcal{Q}^{(\alpha_l,\alpha_l')}_{d_l,d_{l-1}}$ are combined with the left inner index $d_l$ to form the matrix $A_{(d_l,(\alpha_l,\alpha_l')),d_{l-1}}$, whose SVD yields

$$\mathcal{Q}^{(\alpha_l,\alpha_l')}_{d_l,d_{l-1}} = \sum_{j_{l-1}} \overrightarrow{U}_{(d_l,(\alpha_l,\alpha_l')),j_{l-1}} \overrightarrow{\sigma}_{j_{l-1}} \overrightarrow{V}^\dagger_{j_{l-1},d_{l-1}} . \tag{31}$$

Now, the MPO matrix at step $l$ is updated as

$$\tilde{\mathcal{Q}}^{(\alpha_l,\alpha_l')}_{d_l,j_{l-1}} = \overrightarrow{U}_{(d_l,(\alpha_l,\alpha_l')),j_{l-1}} , \tag{32a}$$

and the singular values as well as the matrix $\overrightarrow{V}^\dagger_{j_{l-1},d_{l-1}}$ are passed on the prior step $l-1$

$$\tilde{\mathcal{Q}}^{(\alpha_{l-1},\alpha_{l-1}')}_{j_{l-1},d_{l-2}} = \sum_{d_{l-1}} \overrightarrow{\sigma}_{j_{l-1}} \overrightarrow{V}^\dagger_{j_{l-1},d_{l-1}} \mathcal{Q}^{(\alpha_{l-1},\alpha_{l-1}')}_{d_{l-1},d_{l-2}} . \tag{32b}$$

Analogously to the forward sweep, we cast the update into the shape of a transformation

$$\tilde{\mathcal{Q}}^{(\alpha_l,\alpha_l')}_{d_l,j_{l-1}} = \sum_{d_{l-1}} \mathcal{Q}^{(\alpha_l,\alpha_l')}_{d_l,d_{l-1}} \overrightarrow{T}^{-1}_{d_{l-1},j_{l-1}} , \tag{33a}$$

$$\tilde{\mathcal{Q}}^{(\alpha_{l-1},\alpha_{l-1}')}_{j_{l-1},d_{l-2}} = \sum_{d_{l-1}} \overrightarrow{T}_{j_{l-1},d_{l-1}} \mathcal{Q}^{(\alpha_{l-1},\alpha_{l-1}')}_{d_{l-1},d_{l-2}} , \tag{33b}$$

with

$$\overrightarrow{T}_{j_{l-1},d_{l-1}} = \overrightarrow{\sigma}_{j_{l-1}} \overrightarrow{V}^\dagger_{j_{l-1}d_{l-1}} , \tag{34a}$$

$$\overrightarrow{T}^{-1}_{d_{l-1},j_{l-1}} = \overrightarrow{V}_{d_{l-1}j_{l-1}} \overrightarrow{\sigma}^{-1}_{j_{l-1}} . \tag{34b}$$

The overall transformation matrices in Eq. (20) are modified as

$$\tilde{\mathcal{T}}_{j_l,\beta_l} = \sum_{d_l} \overrightarrow{T}_{j_l,d_l} \mathcal{T}_{d_l,\beta_l} , \tag{35a}$$

$$\tilde{\mathcal{T}}^{-1}_{\beta_l,j_l} = \sum_{d_l} \mathcal{T}^{-1}_{\beta_l,d_l} \overrightarrow{T}^{-1}_{d_l,j_l} . \tag{35b}$$

## 3.3 Overall transformation

We are now in the position to formulate an explicit expression for the transformation matrices in Eq. (20) relating the full environment propagator to the final PT-MPO matrices obtained by the ACE algorithm, which is visualized in Fig. 1(c).

To this end, we concatenate for each of the $N_E$ environment modes the expansion step in Eqs. (24) with the forward and backward sweeps in Eqs. (30) and (35), respectively. Combining

$$\overleftrightarrow{T}_{d_l^{(k)},(d_l^{(k-1)},\beta_l^{(k)})} = \sum_{j_l^{(k)}} \overrightarrow{T}_{d_l^{(k)},j_l^{(k)}} \overleftarrow{T}_{j_l^{(k)},(d_l^{(k-1)},\beta_l^{(k)})} , \tag{36a}$$

$$\overleftrightarrow{T}^{-1}_{(d_l^{(k-1)},\beta_l^{(k)}),d_l^{(k)}} = \sum_{j_l^{(k)}} \overleftarrow{T}^{-1}_{(d_l^{(k-1)},\beta_l^{(k)}),j_l^{(k)}} \overrightarrow{T}^{-1}_{j_l^{(k)},d_l^{(k)}} , \tag{36b}$$

the overall transformation matrices become

$$\mathcal{T}_{d_l^{(N_E)},(\beta_l^{(1)},...,\beta_l^{(N_E)})} = \sum_{d_l^{(1)},...,d_l^{(N_E-1)}} \prod_{k=1}^{N_E} \overleftrightarrow{T}_{d_l^{(k)},(d_l^{(k-1)},\beta_l^{(k)})}, \tag{37a}$$

$$\mathcal{T}^{-1}_{(\beta_l^{(1)},...,\beta_l^{(N_E)}),d_l^{(N_E)}} = \sum_{d_l^{(1)},...,d_l^{(N_E-1)}} \prod_{k=1}^{N_E} \overleftrightarrow{T}^{-1}_{(d_l^{(k-1)},\beta_l^{(k)}),d_l^{(k)}}, \tag{37b}$$

where the index $d_l^{(0)}$ is understood to be restricted to $d_l^{(0)} = 0$, leading to no expansion for the double index $(d_l^{(0)},\beta_l^{(1)}) = \beta_l^{(1)}$.

Several aspects of Eqs. (37) are noteworthy: First, it is instructive to compare the lossy transformation induced by $\mathcal{T}$ and its pseudoinverse $\mathcal{T}^{-1}$ with a conventional projection to a reduced subspace spanned by a set of vectors $\{u_j\}$. The matrix $U$ built by $\{u_j\}$ as column vectors is a truncated unitary with its Moore-Penrose pseudoinverse [54] defined by $U^{-1} = U^\dagger$. Here, however, the presence of the singular values and their reciprocals in Eqs. (29) and (34) leads to transformation matrices and pseudoinverses which are not related by Hermitian conjugation $\mathcal{T}^{-1} \neq \mathcal{T}^\dagger$. In fact, $\mathcal{T}^{-1}$ is not the unique Moore-Penrose pseudoinverse of $\mathcal{T}$, i.e., the column vectors of $\mathcal{T}^{-1}$ do not span the same space as the conjugates of the row vectors of $\mathcal{T}$.

For the purpose of interpretation, this implies that the relevant subspace to which inner bonds of PTs correspond is slightly ambiguous, as it can be identified with the column space of either $\mathcal{T}^{-1}$ or $\mathcal{T}^\dagger$. For practical applications, such as the extraction of information from the inner bonds, it entails that the pseudoinverse $\mathcal{T}^{-1}$ contains additional information and cannot be reconstructed from $\mathcal{T}$ alone.

Moreover, we find from Eq. (37) that the transformation matrices $\mathcal{T}$ and $\mathcal{T}^{-1}$ themselves have the structure of half-open MPOs, where each environment mode $k$ corresponds to a site with outer bond $\beta_l^{(k)}$, while $d_l^{(k)}$ describe the inner bonds. The last inner bond $d_l^{(N_E)}$ of the tranformation matrix MPO remains dangling and is identified the inner bond $d_l$ of the final PT-MPO at step $l$. The dangling bond can be closed by multiplying with $\rho_{d_l}^{\alpha_l}$, the extended density matrix obtained at step $l$ during the ACE algorithm by interation (16). The result

$$\rho_{\alpha_l,\beta_l^{(1)},...,\beta_l^{(N_E)}} = \sum_{d_l^{(N_E)}} \mathcal{T}^{-1}_{(\beta_l^{(1)},...,\beta_l^{(N_E)}),d_l^{(N_E)}} \rho_{d_l^{(N_E)}}^{\alpha_l} = \sum_{d_l^{(1)},...,d_l^{(N_E)}} \left( \prod_{k=1}^{N_E} \overleftrightarrow{T}^{-1}_{(d_l^{(k-1)},\beta_l^{(k)}),d_l^{(k)}} \right) \rho_{d_l^{(N_E)}}^{\alpha_l}, \tag{38}$$

is an MPO representation of the total system and environment density matrix.

## 3.4 Extraction of environment observables

As outlined in Eq. (21) and in line with Eq. (38), the knowledge of $\mathcal{T}^{-1}$ enables the extraction of environment observables as well as mixed system-environment observables. Starting from Eq. (9), the expectation value of a general mixed system environment observable $\hat{O}$ with Liouville space representation $o_{\alpha,\beta}$ given by Eq. (8) is obtained by

$$\langle \hat{O}(t_l) \rangle = \sum_{\alpha_l,\beta_l} o_{\alpha_l,\beta_l} \rho_{\alpha_l,\beta_l} \approx \sum_{\substack{\alpha_l,\beta_l \\ d_l,\beta_l'}} o_{\alpha_l,\beta_l} \mathcal{T}^{-1}_{\beta_l,d_l} \mathcal{T}_{d_l,\beta_l'} \rho_{\alpha_l,\beta_l'} = \sum_{\alpha_l,d_l} \mathfrak{o}_{d_l}^{\alpha_l} \rho_{d_l}^{\alpha_l}, \tag{39}$$

with observable closure

$$\mathfrak{o}_{d_l}^{\alpha_l} = \sum_{\beta_l} o_{\alpha_l,\beta_l} \mathcal{T}^{-1}_{\beta_l,d_l}. \tag{40}$$

Here, two issues arise if Eqs. (39) and (40) are to be used to extract environment observables in practice: First, the matrices $\mathcal{T}^{-1}_{\beta_l,d_l}$ change from time step to time step. Even though they can be represented in MPO form by Eq. (37b), storing them for all time steps $l$ requires large amounts of memory. This can be circumvented by fixing an operator $\hat{O}$ at the start and updating the corresponding $\mathfrak{o}^{\alpha_l}_{d_l}$ in every expansion and compression step of the ACE algorithm in the same way that $\mathcal{T}^{-1}_{\beta_l,d_l}$ would be updated, namely by multiplying with $\delta_{(d_l,\beta_l^{(K)}),\tilde{d}_l}$ as in Eq. (24b), $\overleftarrow{T}^{-1}_{d_l,j_l}$ as in Eq. (30b), and $\overrightarrow{T}^{-1}_{d_l,j_l}$ as in Eq. (35b).

The second issue is that $o_{\alpha_l,\beta_l}$ itself can be unwieldy in general, as $\beta_l$ runs over a complete basis set for the full environment Liouville space. Here, we therefore limit the discussion to operators of the form

$$\hat{O} = \sum_{k=1}^{N_E} \hat{A} \otimes \hat{O}^{(k)}, \tag{41}$$

where $\hat{A}$ acts on the system Hilbert space $\mathcal{H}_S$ and $\hat{O}^{(k)}$ acts on the $k$-th environment mode. We define the corresponding Liouville operators as

$$o_{\alpha,\beta^{(1)},\ldots,\beta^{(N_E)}} = o^\alpha \sum_{k=1}^{N_E} o^{(k)}_{\beta^{(k)}}. \tag{42}$$

Then, the corresponding observable closure $\mathfrak{o}^{\alpha_l}_{d_l}$ defined in Eq. (40) is obtained alongside the PT in the ACE algorithm by the iteration

$$\mathfrak{o}_{d_l^{(k)}} = \sum_{d_l^{(k-1)}} \sum_{\beta_l^{(k)}} \left( \mathfrak{o}_{d_l^{(k-1)}} \mathfrak{I}_{\beta_l^{(k)}} + q_{d_l^{(k-1)}} o^{(k)}_{\beta^{(k)}} \right) \overleftarrow{T}^{-1}_{(d_l^{(k-1)},\beta_l^{(k)}),d_l^{(k)}}, \tag{43}$$

where the first summand describes the expansion and transformation of $\mathfrak{o}_{d_l^{(k-1)}}$, which accounts for contributions to the overall observable from modes $1,2,\ldots,k-1$, while the second summand adds the contribution $o^{(k)}_{\beta^{(k)}}$ from environment mode $k$ corresponding to the $k$-th term of the sum in Eq. (42). The symbol $\mathfrak{I}_{\beta_l^{(k)}}$ accounts for the trace over the $k$-th environment mode as introduced below Eq. (10), and $q_{d_l^{(k)}}$ are the closures at intermediate time steps corresponding to a trace over environment modes $1,2,\ldots k$, which are also obtained iteratively by

$$q_{d_l^{(k)}} = \sum_{d_l^{(k-1)}} \sum_{\beta_l^{(k)}} q_{d_l^{(k-1)}} \mathfrak{I}_{\beta_l^{(k)}} \overrightarrow{T}^{-1}_{(d_l^{(k-1)},\beta_l^{(k)}),d_l^{(k)}}. \tag{44}$$

The final observable closure $\mathfrak{o}^{\alpha_l}_{d_l}$ is identified with the result of the iteration (43) multiplied with the system part of the operator $\mathfrak{o}^{\alpha_l}_{d_l} = o^{\alpha_l} \mathfrak{o}_{d_l^{(N_E)}}$. Moreover, identifying $q_{d_l} = q_{d_l^{(N_E)}}$ provides an alternative way to that described in Ref. [23] for obtaining trace closures $q_{d_l}$ used to extract the reduced system density matrix from PT-MPO calculation at intermediate time steps via $\bar{\rho}_{\alpha_l} = \sum_{d_l} q_{d_l} \rho^{\alpha_l}_{d_l}$.

## 4 Examples

### 4.1 Proof of principle: Environment spins

First, we demonstrate a minimal example for the extraction of environment observables on the basis of a central spin model. We consider a central spin $\mathbf{S}$ coupled to a bath of $N_E$ spins $\mathbf{s}^{(k)}$

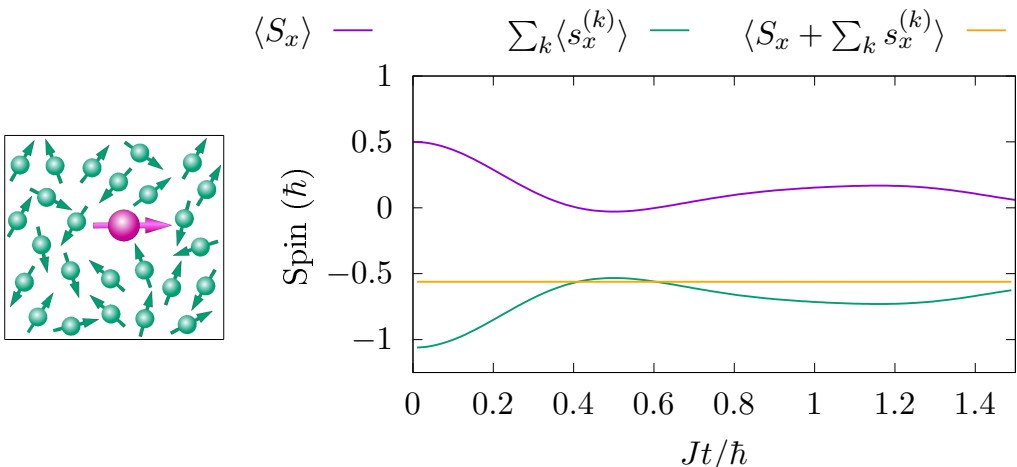

Figure 2: Dephasing of a central spin $S_x$, initially polarized along the $x$ axis, in an ensemble of $N_E = 25$ environment spins, initially randomly oriented (see sketch on the left). The overall environment spin projection $\sum_k s_x^{(k)}$ is extracted from the inner degrees of freedom of the PT-MPO. The conservation of the total spin $S_x + \sum_k s_x^{(k)}$ is correctly reproduced.

via the Heisenberg interaction

$$H_E = \sum_{k=1}^{N_E} J\, \mathbf{S} \cdot \mathbf{s}^{(k)}. \tag{45}$$

Here, we assume that the central spin is initially polarized along the $x$ direction $S_x(t=0) = \frac{1}{2}$, $S_y(t=0) = S_z(t=0) = 0$, while the $N_E = 25$ environment spins are initially randomly oriented. The central spin $\mathbf{S}$ is obtained as usual with ACE from the reduced system density matrix, while the sum of environment spins $\sum_k \mathbf{s}^{(k)}$ is obtained as an environment observable by tracking the corresponding observable closure during MPO compression. The PT-MPO is calculated for convergence parameters $(J/\hbar)\Delta t = 0.01$ and $\epsilon = 10^{-11}$, for which the full environment Liouville space dimension $4^{25} \approx 10^{15}$ is compressed to a maximal bond dimension of 136.

The ensuing spin dynamics ($x$-component) is depicted in Fig. 2. The central spin $S_x$ dephases in the bath of environment spins, yet the dephasing is incomplete due to the finite size of the spin bath. The dynamics of the collective environment spin $\sum_k s_x^{(k)}$ changes accordingly as the total central plus environment spin $S_x + \sum_k s_x^{(k)}$ is conserved by the Hamiltonian. This is correctly reproduced in Fig. 2 by our numerical approach demonstrating the successful extraction of environment observables.

## 4.2 Mixed system-environment observables: Currents

To infer physical insights from measurement of currents through zero-dimensional quantum structures like single molecules or quantum dots, a comparison with theoretical predictions is highly desirable [55]. A typical scenario for charge transport involves a single quantum site, which is described by a two-level system (the site being either occupied or not), and which is coupled to two metallic leads at different chemical potentials $\mu_1 > \mu_2$. The leads are modelled as fermionic baths. The total Hamiltonian is $H = H_S + H_{E_1} + H_{E_2}$ with $H_{E_i} = \sum_k H_{E_i}^{(k)}$ and

$$H_{E_i}^{(k)} = \hbar\omega_{i,k} c_{i,k}^\dagger c_{i,k} + \hbar g_{i,k}\left(c_{i,k}^\dagger c_S + c_S^\dagger c_{i,k}\right), \tag{46}$$

where $c_{i,k}^{(\dagger)}$ denote fermionic annihilation (creation) operators for the $k$-th mode of the $i$-th environment, $\hbar\omega_{i,k}$ and $g_{i,k}$ are the corresponding energies and coupling constants, and $c_S^\dagger$ and $c_S$ describe fermionic operators of the system mode. We set $H_S = 0$, and assume that the central site is initially unoccupied, the first lead is completely filled and the second lead is completely empty (chemical potentials $\mu_{1/2} = \pm\infty$). The energies and couplings for both baths are obtained by uniformly discretizing the spectral densities $J_i(\omega) = \sum_k g_{i,k}^2 \delta(\omega - \omega_{i,k})$ of the shape of a bump function

$$J_i(\omega) = \begin{cases} \frac{\kappa}{2\pi}\exp\left(1 - \frac{1}{1-(2\omega/\omega_{BW})^2}\right), & |\omega| < \frac{1}{2}\omega_{BW}, \\ 0, & \text{else.} \end{cases} \tag{47}$$

This is a smooth function with compact support on an interval with bandwidth $\omega_{BW}$ (see inset in Fig. 3).

The particle current flowing from the central site into lead $i$ is equal to the change of the total occupations of the lead due to coupling to the site. From the Heisenberg equations of motion we find

$$I_i := \frac{\partial}{\partial t}\sum_k \langle c_{i,k}^\dagger c_{i,k}\rangle = \frac{i}{\hbar}\sum_k \langle [H, c_{i,k}^\dagger c_{i,k}]\rangle = 2\sum_k g_{i,k}\text{Im}\left\{\langle c_{i,k}^\dagger c_S\rangle\right\}. \tag{48}$$

This is a mixed system-environment observable, which depends on correlations between the system and the environment. At first glance, this observable as well as the environment Hamiltonian in Eq. (46) seem decomposable into sums of terms each involving only creation and annihilation operators of a single environment mode, as we require for the environment observable extraction described in this article. Note, however, that fermionic operators obey the canonical anticommutator relations, whereas the decomposition into independent modes so far implicitly assumed that operators for different modes commute. As described in more detail in Ref. [53], PT-MPOs obeying the proper fermionic anticommutator relations can be obtained by a slight modification of ACE based on a Jordan-Wigner transformation. Here, we briefly summarize the main aspects while using a formulation that facilitates the extraction of the current $I_1(t)$ in a setup with up to two environments.

First, the Jordan-Wigner transformations requires an ordering of fermionic modes. Here, we assume the order

$$(1,1), (1,2), \ldots, (1, N_{E_1}), (S), (2, N_{E_2}), \ldots, (2,2), (2,1), \tag{49}$$

where $(S)$ denotes the system mode and $(i, k)$ denotes the $k$-th mode of the $i$-th environment. The anticommutation relations remain fulfilled when the fermionic operators are replaced by spin-1/2 climbing operators

$$c_{i,k} = P^{[(1,1),(i,k-1)]}\sigma_{i,k}^-, \tag{50a}$$

$$c_{i,k}^\dagger = P^{[(1,1),(i,k-1)]}\sigma_{i,k}^+, \tag{50b}$$

where

$$P^{[(i_1,k_1),(i_2,k_2)]} = \prod_{(i,k)=(i_1,k_1)}^{(i_2,k_2)}(-\sigma_{i,k}^z), \tag{51}$$

is the product of parity operators from mode $(i_1, k_1)$ to $(i_2, k_2)$ in the order given by Eq. (49), which takes the value +1 (-1) if the modes in the range have an even (odd) number of occupations. Applying this transformation to the Hamiltonian in Eq. (46), one can show that

$$H_{E_i}^{(k)} = \left(P_1^{[(1,1),(i,k-1)]}\right)^{\sigma_S^+\sigma_S^-}\tilde{H}_{E_i}^{(k)}\left(P_1^{[(1,1),(i,k-1)]}\right)^{\sigma_S^+\sigma_S^-}, \tag{52}$$

where

$$\tilde{H}_{E_i}^{(k)} = \hbar\omega_{i,k}\sigma_{i,k}^+\sigma_{i,k}^- + \hbar g_{i,k}\big(\sigma_{i,k}^+\sigma_S^- + \sigma_S^+\sigma_{i,k}^-\big), \tag{53}$$

is the spin analogue of the fermionic Hamiltonian $H_{E_i}^{(k)}$.

Now, while the fermionic mode Hamiltonians individually contain non-local terms via the parity operators, it can be shown that these non-local terms cancel when the propagators of all modes are combined, leaving only local parity terms [53]. Specifically, representing the PT-MPO matrices for environment $i$ using lossy compression matrices as well as the symmetric Trotter decomposition of the full environment propagator in Eq. (19), one finds

$$\begin{aligned}
\mathcal{Q}^{[i]} &= \mathcal{T}e^{\mathcal{L}_{E_i}^{(N_{E_i})}\frac{\Delta t}{2}}\dots e^{\mathcal{L}_{E_i}^{(2)}\frac{\Delta t}{2}}e^{\mathcal{L}_{E_i}^{(1)}\Delta t}\mathcal{L}_{E_i}^{(2)\frac{\Delta t}{2}}\dots e^{\mathcal{L}_{E_i}^{(N_{E_i})}\frac{\Delta t}{2}}\mathcal{T}^{-1} \\
&= \mathcal{T}\mathcal{B}^{[i,N_{E_i}]}\dots\mathcal{B}^{[i,2]}\mathcal{B}^{[i,1]}\mathcal{B}^{[i,1]}\mathcal{B}^{[i,2]}\dots\mathcal{B}^{[i,N_{E_i}]}\mathcal{T}^{-1},
\end{aligned} \tag{54}$$

where

$$\mathcal{B}^{[i,k]} := e^{\tilde{\mathcal{L}}_{E_i}^{(k)}\frac{\Delta t}{2}}\Big[\big(-\sigma_{i,k}^z\big)^{\sigma_S^+\sigma_S^-}\otimes\big(-\sigma_{i,k}^z\big)^{\sigma_S^+\sigma_S^-}\Big], \tag{55}$$

are the local environment mode propagators corresponding to the spin Hamiltonians $\tilde{H}_{E_i}^{(k)}$ in Eq. (53) modified by local parity operators $\big(-\sigma_{i,k}^z\big)^{\sigma_S^+\sigma_S^-}$. To summarize, PT-MPOs correctly accounting for fermionic anticommutation can be obtained simply by replacing the individual mode propagators $\mathcal{B}$ in Eq. (22) by corresponding $\mathcal{B}^{[i,k]}$ in Eq. (55).

To extract the particle current via the inner bonds of the fermionic PT-MPO, we apply the Jordan-Wigner transform to Eq. (48), which yields

$$I_i := 2\sum_k g_{1,k}\text{Im}\left\{\langle\sigma_{i,k}^+ P_1^{[k+1,N_{E_i}]}\sigma_S^-\rangle\right\}. \tag{56}$$

Again the non-local parity terms make adaptations necessary. This is achieved by replacing the iteration in Eq. (43) by

$$\mathfrak{o}_{d_l^{(k)}} = \sum_{d_l^{(k-1)}}\sum_{\beta_l^{(k)},\xi,\eta}\delta_{\beta_l^{(k)},(\xi,\eta)}\Big(\mathfrak{o}_{d_l^{(k-1)}}\langle\eta|-\sigma^z|\xi\rangle + q_{d_l^{(k-1)}}\langle\eta|\sigma^+|\xi\rangle\Big)\overleftrightarrow{T}^{-1}_{(d_l^{(k-1)},\beta_l^{(k)}),d_l^{(k)}}. \tag{57}$$

With this observable closure, we are now in the position to extract the fermionic particle current $I_i(t)$ as a mixed system-environment observable from the inner bonds of the fermionic PT-MPO.

Finally, we point out that the numerically exact simulation in the presence of two generally non-Markovian environments is possible by propagating the time evolution with two PT-MPOs, which are calculated independently of each other. This was utilized already for bosonic environments, e.g., in Refs. [17, 37]. Here, the ordering used for the Jordan-Wigner transformation in Eq. (49) ensures that no additional non-local parity operators coupling the two environments emerge. The fact that this treatment remains numerically exact can be seen straightforwardly in the notation developed here: Using transformation matrices $\mathcal{T}^{[i]}$ and PT-MPO matrices $\mathcal{Q}^{[i]} = \mathcal{T}^{[i]}e^{\mathcal{L}_{E_i}\Delta t}\big(\mathcal{T}^{[i]}\big)^{-1}$ for bath $i$, we find that

$$\begin{aligned}
\mathcal{Q}^{[1]}\mathcal{Q}^{[2]} &= \mathcal{T}^{[1]}\mathcal{T}^{[2]}e^{\mathcal{L}_{E_1}\Delta t}e^{\mathcal{L}_{E_2}\Delta t}\big(\mathcal{T}^{[2]}\big)^{-1}\big(\mathcal{T}^{[1]}\big)^{-1} \\
&= \mathcal{T}e^{(\mathcal{L}_{E_1}+\mathcal{L}_{E_2})\Delta t}\mathcal{T}^{-1} + \mathcal{O}(\Delta t^2),
\end{aligned} \tag{58}$$

describes the joint influence of both environments up to a controlled Trotter error. Note that the order of the Trotter error can be reduced when alternating the order of multiplication

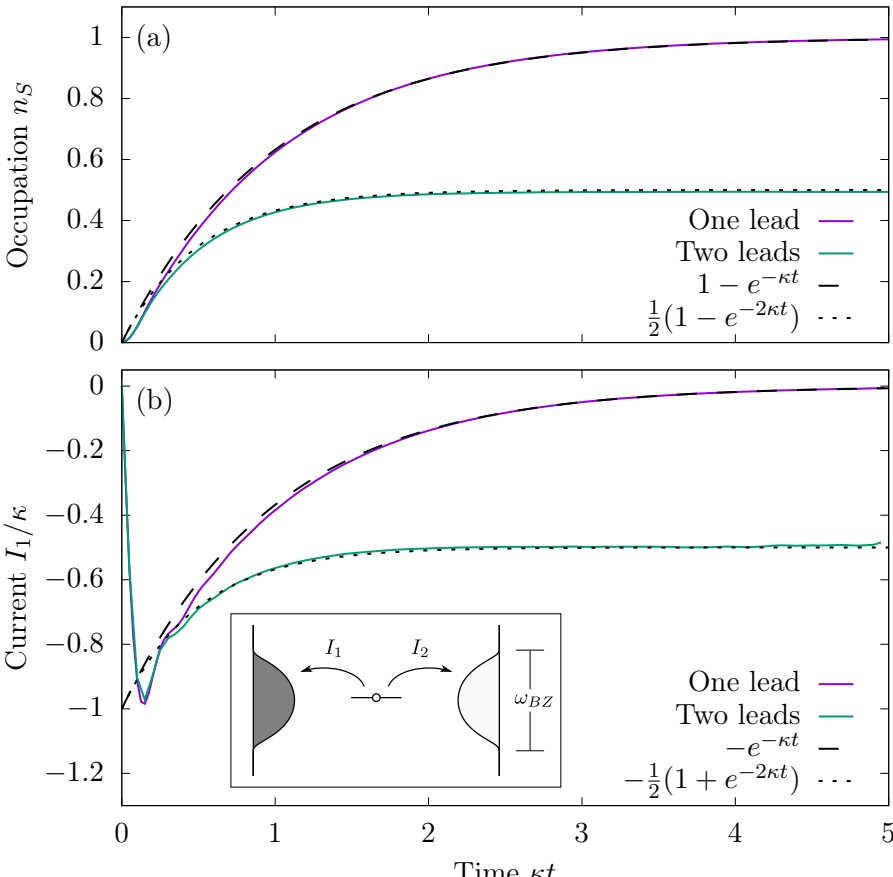

Figure 3: Occupations (a) and currents (b) in charge transport between a single site and one or two metallic leads at chemical potentials $\mu_{1/2} = \pm\infty$, as depicted in the inset of (b). Dashed and dotted black lines depict results in the Markov limit for the single lead case and the two lead case, respectively.

(with respect to the system indices) for subsequent time steps, i.e. setting $\mathcal{Q} = \mathcal{Q}^{[1]}\mathcal{Q}^{[2]}$ for odd and $\mathcal{Q} = \mathcal{Q}^{[2]}\mathcal{Q}^{[1]}$ for even times steps, because this amounts to a symmetric Trotter decomposition of a propagation over two time steps. For a consistent accuracy, observables at odd times steps are disregarded.

In Fig. 3(a) and (b) we show the system site occupations and the particle current from the system to the first lead, respectively, for simulations involving either only the first lead or both leads. The PT-MPOs are calculated using parameters $\omega_{BW} = 64\kappa$, $N = 128$, $\Delta t = 0.025/\kappa$, and $\epsilon = 10^{-7}$.

In the case of a single lead with large chemical potential, carriers flow from lead to the system site (negative $I_1$) until the latter is fully populated. The dynamics is largely in agreement with the analytical behavior $n_S(t) = 1 - e^{-\kappa t}$ in the Markov limit, which is expected to hold as the bath correlation time $\sim 1/\omega_{BW}$ is short compared to the relaxation time $1/\kappa$. Due to the conservation of particle number and the coupling to only a single lead, the current can be obtained from the system evolution via $I_1(t) = -\frac{\partial}{\partial}n_S(t)$, which is $I_1(t) = -\kappa e^{-\kappa t}$ in the Markov limit.

For a site coupled to two leads, changes of site occupations are due to currents to either lead, which generally obfuscates the relation between currents through one of the leads and system observables. This is where extracting the current via the mixed system-environment observable defined in Eq. (48) is useful. In the present scenario where initially one lead is fully oc-

cupied and one lead is completely empty, the dynamics can be compared to its Markovian limit, where it is governed by the rate equation $\frac{\partial}{\partial t} n_S = -I_1(t) - I_2(t)$, where $I_1(t) = \kappa(n_S(t) - 1)$ and $I_2(t) = \kappa n_S(t)$ are the Markovian currents to environment 1 and 2, respectively. This equation of motion is solved by system occupations $n_S(t) = \frac{1}{2}(1 - e^{-2\kappa t})$, from which we get the current $I_1(t) = -\frac{\kappa}{2}(1 + e^{-2\kappa t})$. As can be seen in Fig. 3, for the case of two leads, simulated site occupations as well as currents also match the Markovian prediction well after a short initial phase on the timescale $\sim 1/\omega_{BW}$.

Summarizing, we have demonstrated that inner bonds of PT-MPOs can be used to extract mixed system-environment operators, which facilitate, e.g., the analysis of currents. In principle, the combination of multiple PT-MPOs also makes it possible to address more complex questions, such as the impact of strong phonon coupling on charge and excitation transport. As this topic requires a more detailed analysis, we leave it for future work.

### 4.3 Convergence of different observables: Application in photon emission

Because the transformation matrices $\mathcal{T}$ are lossy, i.e. rank-reducing, there is no *a priori* guarantee that a given environment observable can be extracted faithfully by the corresponding pseudoinverse $\mathcal{T}^{-1}$. Moreover, the accuracies may be different for different environment observables. This fact can be utilized to probe the information content of inner bonds of PT-MPOs by numerically testing the convergence of different environment observables as a function of the parameters controlling MPO compression. The insights gained by this process then helps to identify alternative ways to extract environment observables with a higher degree of accuracy.

We explore this on the example of radiative decay from a two-level quantum emitter. The light-matter interaction is given by

$$H_E = \sum_k \hbar\omega_k a_k^\dagger a_k + \sum_k \hbar g_k\big(a_k^\dagger |g\rangle\langle e| + a_k |e\rangle\langle g|\big), \tag{59}$$

where $a_k^\dagger$ and $a_k$ are photon creation and annihilation operators and $|g\rangle$ as well as $|e\rangle$ are ground and excited states of the emitter, respectively. Photon mode energies $\hbar\omega_k$ are chosen to uniformly discretize an interval $[-\hbar\omega_{BW}/2, \hbar\omega_{BW}/2]$ with frequency bandwidth $\omega_{BW}$ using $N_E$ modes. The couplings are obtained by $g_k = \sqrt{J(\omega_k)\omega_{BW}/N_E}$ with a flat spectral density $J(\omega) = \kappa/(2\pi)$ [see Fig. 4(a)]. For a large enough bandwidth $\omega_{BW} \gg \kappa$, the photon environment is Markovian [23] and describes radiative decay of the emitter excitation $n_e = \langle(|e\rangle\langle e|)\rangle$ with rate $\kappa$. Thus, for an initially excited emitter $n_e(0) = 1$ the excitation decays as $n_e^{\mathrm{Markov}} \approx e^{-\kappa t}$. As the Hamiltonian conserves the total number of excitations, it follows that the number of photons emitted into the environment $n_{\mathrm{ph}} = \sum_k \langle a_k^\dagger a_k\rangle$ is $n_{\mathrm{ph}}^{\mathrm{Markov}} = 1 - e^{-\kappa t}$.

We use this test case to assess the convergence of the two-level system excitations $n_e$ as well as the environment observable $n_{\mathrm{ph}}$. The numerically calculated dynamics is depicted in Fig. 4(b) and (c) for different MPO compression thresholds $\epsilon$, fixed bandwidth $\omega_{BW} = 200\kappa$, number of modes $N_E = 400$, and time discretization $\Delta t = 0.05/\kappa$.

The system observable $n_e$ converges quickly and becomes virtually indistinguishable from the Markovian result for thresholds $\epsilon = 10^{-5}$ and smaller. In contrast, the extraction of the photon number $n_{\mathrm{ph}}$ as an environment observable is found to be more unstable and converges much more slowly. This is because the compression of the PT-MPO in ACE is only designed to accurately reproduce the reduced system density matrix and, hence, system observables. If the compression is optimal in this sense, the accuracy of the extracted environment observables should be determined by how important they are for influencing system observables in future time steps.

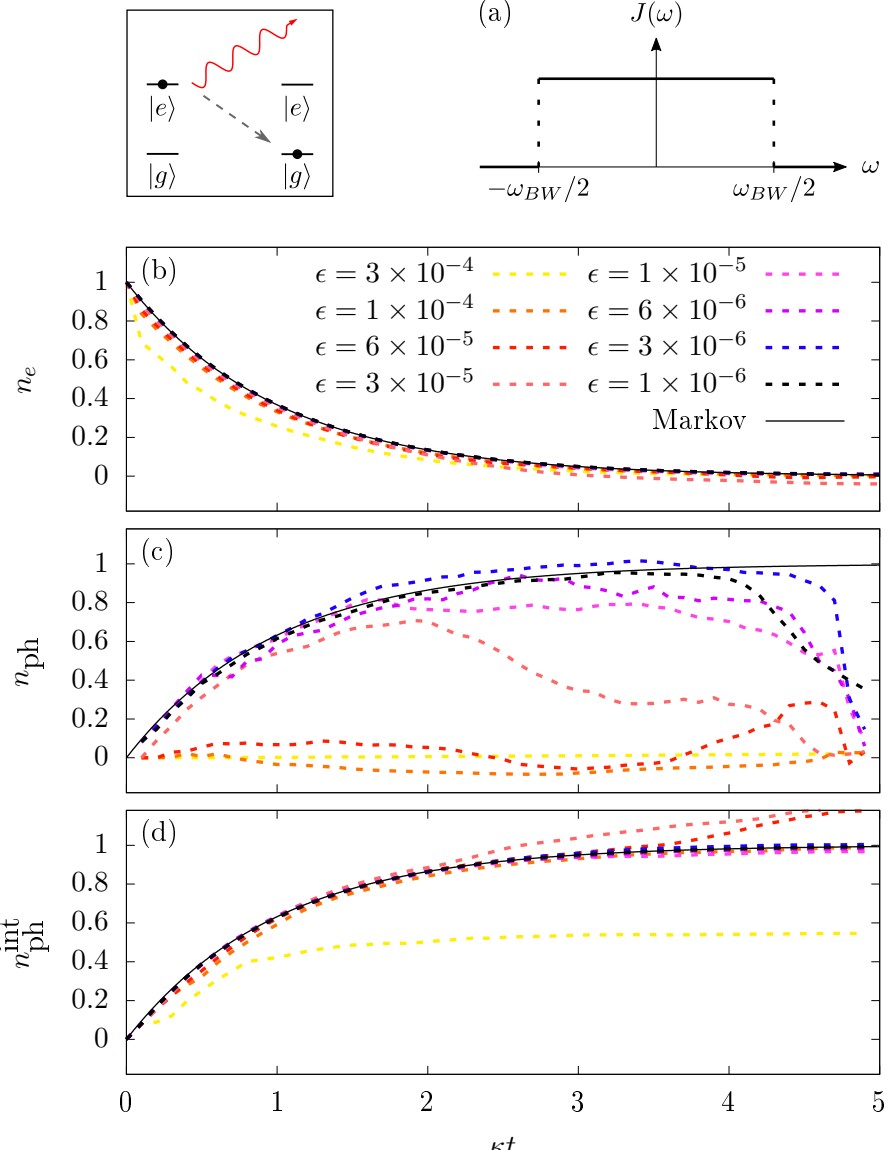

Figure 4: Photon emission from an initially excited two-level system (see sketch). This is described microscopically by a multi-mode Jaynes-Cummings model with flat spectral density over a bandwidth $\omega_{BW}$ as depicted in (a). Two-level system occupation $n_e$ (b) and emitted photon number $n_{\text{ph}}$, where the photon number is either extracted directly (c) or by integrating its equation of motion, whose driving term is obtained by extracting system-environment correlations (d) as in Eq. (62).

To test this hypothesis, we consider the Heisenberg equations of motion for emitter occupations

$$\frac{\partial}{\partial t} n_e = \frac{i}{\hbar} \langle [H_S + H_E, |e\rangle\langle e|] \rangle$$
$$= \frac{i}{\hbar} \langle [H_S, |e\rangle\langle e|] \rangle + 2 \sum_k g_k \text{Im}\{\langle (|e\rangle\langle g|a_k) \rangle\}. \tag{60}$$

Note that the system evolution is directly driven by system-environment correlations $\langle (|e\rangle\langle g|a_k) \rangle$ but not by the photon number $n_{\text{ph}}$ itself. The latter affects the system only indirectly by influencing the evolution of the system-environment correlations. Because a finite

time is needed for the influence of $n_{\mathrm{ph}}$ on the correlations to result in a measurable effect on the system, $n_{\mathrm{ph}}$ is reproduced most accurately in Fig. 4(c) at earlier points in time. In contrast, $n_{\mathrm{ph}}$ at later time steps can no longer affect the state of the system within the remaining propagated time in the simulation, so the extracted value of $n_{\mathrm{ph}}$ at the last few time steps remains unreliable even for small thresholds $\epsilon = 10^{-6}$.

This explanation also provides a hint on how to construct a more accurate scheme to extract the total emitted photon number: Considering the equation of motion

$$\frac{\partial}{\partial t} n_{\mathrm{ph}} = \frac{i}{\hbar} \langle [H_E, \sum_k a_k^\dagger a_k] \rangle = -2 \sum_k g_k \mathrm{Im} \{ \langle (|e\rangle\langle g| a_k) \rangle \} , \tag{61}$$

we find that the photon number is driven by the same system-environment correlations as the emitter occupation. As these correlations influence the system directly, they are likely better reproduced when extracted from the inner bonds of the PT-MPO than $n_{\mathrm{ph}}$ itself. This suggests obtaining the photon number by integrating Eq. (61)

$$n_{\mathrm{ph}}^{\mathrm{int}}(t) = \int_0^t d\tau \Big( \frac{\partial}{\partial \tau} n_{\mathrm{ph}}(\tau) \Big) = \int_0^t d\tau \sum_k g_k \langle (|e\rangle\langle g| a_k) \rangle_\tau , \tag{62}$$

where $\langle (|e\rangle\langle g| a_k) \rangle_\tau$ are the system-environment correlations at time $\tau$, which are extracted via the corresponding observable closures.

The result of this approach is depicted in Fig. 4(d). Indeed, the convergence with respect to the compression threshold $\epsilon$ is much faster compared to the direct extraction of $n_{\mathrm{ph}}$ in Fig. 4(c). This corroborates the argument that the inner bonds of PT-MPOs convey more information about first-order system-environment correlations that directly affect the system dynamics compared to environment observables with a more indirect influence on the system. The hierarchy of Heisenberg equations of motion starting from system observables provide a useful basis for a qualitative estimation of the expected accuracy of environment observable extraction via inner bonds of PT-MPOs.

The nearly Markovian environment with a flat spectral density provided an ideal test case for different observable extraction schemes due to the availability of analytical solutions. Numerically exact open quantum systems approaches are most useful in cases of non-Markovian environments, where no analytical solutions exist. If these systems are also externally driven, inferring environment observables from conservation laws is often no longer possible. We now show how the approach laid out above can be applied to investigate photon emission in a model of a two-level emitter coupled to a structured, non-Markovian photonic environment with spectral density given by a Lorentzian [see Fig. 5(a)]

$$J(\omega) = \frac{\kappa^2}{\pi} \frac{\gamma}{\omega^2 + \gamma^2} , \tag{63}$$

where $\kappa$ determines the overall interaction strength and $\gamma$ denotes the width of the Lorentzian. The corresponding PT-MPO is calculated restricting the spectral density to a frequency interval $\omega_{BW} = 200\kappa$, which is discretized by $N_E = 2000$ photon mode with a Hilbert space containing up to 2 photons per mode. Furthermore, we use parameters $\gamma = \frac{1}{4}\kappa$, $\Delta t = 0.05/\kappa$, and $\epsilon = 10^{-7}$.

In Fig. 5(b), we show the free ($H_S = 0$) emission dynamics of an initially prepared emitter excitation. The strongly peaked structure in the environment results in underdamped oscillations of emitter population $n_e$. The emitted photon number $n_{\mathrm{ph}}^{\mathrm{int}}$ extracted by integrating the system-emitter correlations as in Eq. (62) mirrors the behavior of the emitter population, since the light-matter coupling conserves the total number of excitations.

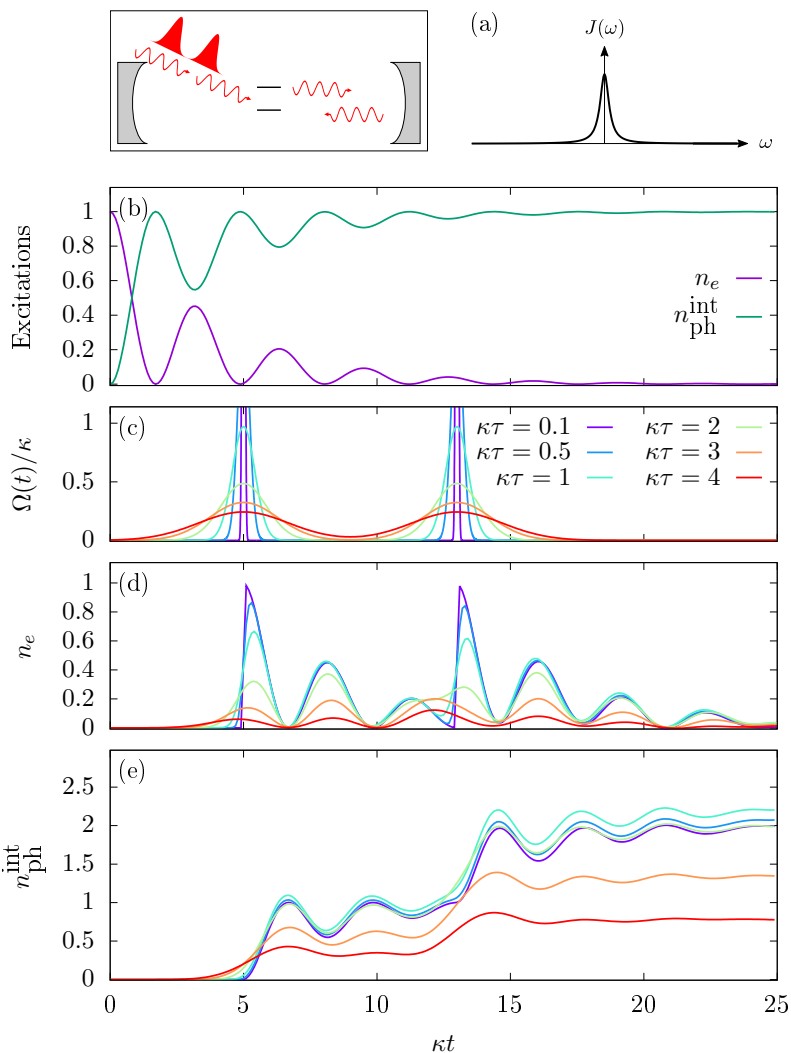

Figure 5: Radiative decay from a two-level quantum emitter into a structured photon environment such as a lossy single-mode microcavity (see sketch), which is modeled with a Lorentzian spectral density $J(\omega)$ (a). (b) Free radiative decay of initially prepared emitter populations $n_e$. The photon number $n_{\text{ph}}^{\text{int}}$ is extracted by integrating system-environment correlations as in Eq. (62). (d) Emitter populations and (e) emitted photon number for the same system driven by two Gaussian pulses depicted in (c). Colors correspond to different pulse widths $\tau_1 = \tau_2 = \tau$.

Next, we drive the two-level system, initially in its ground state, with a sequence of two resonant Gaussian pulses

$$H_S = \frac{\hbar}{2}\Omega(t)\big(|e\rangle\langle g| + |g\rangle\langle e|\big),\tag{64}$$

with

$$\Omega(t) = \frac{A_1}{2\pi\sigma_1}e^{-\frac{1}{2}(t-t_1)^2/\sigma_1^2} + \frac{A_2}{2\pi\sigma_2}e^{-\frac{1}{2}(t-t_2)^2/\sigma_2^2},\tag{65}$$

where $A_1 = A_2 = \pi$ are the pulse areas, $t_1 = 5/\kappa$ and $t_2 = 13/\kappa$ are the centers of the pulses, and $\sigma_i = \tau_i/\sqrt{8\ln 2}$ are the standard deviations with FWHM pulse durations $\tau_i$. The time-dependent Rabi frequency $\Omega(t)$ is depicted in Fig. 5(c) for different values of the pulse widths $\tau_1 = \tau_2 = \tau$.

The corresponding two-level excitations $n_e$ and photon numbers $n_{\text{ph}}^{\text{int}}$ are shown in Fig. 5(d) and (e), respectively. For short pulses with $\kappa\tau = 0.1$, the two-level excitation $n_e$ after the first pulse closely resembles the dynamics of the initial value calculation in Fig. 5(b). The second pulse hits the two-level system when it is nearly in its ground state, promoting it again to the excited state, for which the excitation is again transferred to the photonic environment with a dynamics similar to an initial value calculation. Thus, each pulse introduces one excitation that is eventually converted to a photon, so the photon number $n_{\text{ph}}^{\text{int}}$ approaches a value of 2.

For longer pulses, we first observe an increase in the final photon number $n_{\text{ph}}^{\text{int}}$. This is explained by the fact that excitations are emitted as photons already within the pulse duration $\tau$, which facilitates the extraction of more than one excitation per laser pulse. For even longer pulses, when the pulse width becomes comparable to the period of excitation oscillations between system and environment, excitation oscillations generated at different points in time within the pulse duration destructively interfer. As a result, the capabilities of the composite emitter and environment system to absorb excitations from the external pulse are reduced, leading to values of $n_{\text{ph}}^{\text{int}}$ well below 2.

## 4.4 Inner bonds of PT-MPOs from different algorithms: Example in quantum thermodynamics

The construction of environment observable closures relies on the key property of the ACE algorithm that the relation between inner bonds and the Liouville space of individual environment modes is clearly identifiable before MPO compression. Other algorithms for constructing PT-MPOs [25,29,32] are based on expressions for the Feynman-Vernon influence functional [19], where the environment degrees of freedom have been integrated out using path integral techniques. This obfuscates the relation between the inner bonds of the PT-MPOs and the space of environment excitations. As the starting point of these algorithms is the bath correlation function, they make no reference to any particular discretization of the environment mode continuum. This raises the question whether our interpretation of inner PT-MPO bonds is specific to ACE or applies more universally to any PT-MPO technique.

One argument in favour of the latter is the fact that the physical influence of the environment, and thus the Feynman-Vernon influence functional, should be identical for all methods that are numerically exact. Moreover, MPO compression using SVDs provides at least locally optimal compression [21]. If the PT-MPO algorithms under consideration were to yield globally optimal compression, one would expect the resulting PT-MPOs to be identical, up to the intrinsic gauge freedom of MPOs. On the other hand, comparisons have already revealed that different algorithms may lead to PT-MPOs with different inner bond dimensions, e.g., due to accumulation of numerical error or truncation of MPOs in non-normalized form [29].

Here, we address this question in a numerical experiment. We calculate PT-MPOs for an open quantum system first using ACE and then using the algorithm by Jørgensen and Pollock (JP) [25], which is based on the Feynman-Veron influence functional. We calculate environment observable closures for the ACE PT-MPO and, after fixing the gauges, apply these closures to the inner bonds of PT-MPOs obtained from the JP algorithm. If the information conveyed in the inner bonds of PT-MPOs is universal, i.e. independent of the details of the algorithm, we expect to find at least qualitative agreement between environment observables extracted from both PT-MPOs.

To remove ambiguities due to the gauge freedom, we proceed as follows: After calculating PT-MPOs using the two algorithms, we perform additional sweeps to ensure that the PT-MPOs are stationary. A final forward sweep implicitly orders the basis of the inner bonds such that the rows of the PT-MPO matrices with respect to the inner indices are associated to singular values in decreasing order. Whenever for a give time step the inner dimensions of the PT-

MPOs calculated using both approaches differ, the observable closures from ACE calculations are truncated or zero-padded to match the bond dimensions of the JP PT-MPO. A remaining ambiguity arises from common phase factors within each row of the PT-MPO matrices, which may be passed on to the corresponding columns of the PT-MPO matrix of the next time step. We fix this by extracting the phase of the largest element within each row of an ACE PT-MPO matrix and modify the corresponding phase of the JP PT-MPO.

We apply this approach to a test case in quantum thermodynamics [56–60]. In this research area, one investigates how concepts of macroscopic thermodynamics can be generalized to quantum mechanical systems. While work and heat play a central role in classical thermodynamics, their definition is more complex in quantum settings [61, 62]. Irrespective of such subtleties we are here interested in calculating the total energy absorbed by a quantum system subject to external driving. If this system is a non-Markovian open quantum system, a further challenge is to resolve how the total absorbed energy is distributed over different terms in the Hamiltonian, namely the mean system energy, the mean energy absorbed into purely environmental degrees of freedom, and the mean system-environment interaction energy. The latter is a consequence of non-vanishing system-environment correlations and highlights the need for methods in quantum thermodynamics that can account for strong system-environment coupling [56–60]. Here, the energy distribution over the different terms can be readily obtained from PT-MPOs via their inner bonds.

Concretely, we consider a two-level quantum emitter in contact with a phonon bath described by the spin-boson model.

$$H = H_S + H_E^0 + H_I + H_{PS} \,, \tag{66a}$$

$$H_E^0 = \sum_k \hbar \omega_k b_k^\dagger b_k \,, \tag{66b}$$

$$H_I = \sum_k \hbar g_k (b_k^\dagger + b_k) |e\rangle\langle e| \,, \tag{66c}$$

$$H_{PS} = \sum_k \hbar \frac{g_k^2}{\omega_k} |e\rangle\langle e| \,, \tag{66d}$$

where $H_S$ is Hamiltonian acting only on the two-level system, $H_E^0$ is the energy of the free phonon path, $H_I$ describes the system-environment interaction, and $H_{PS}$ is added to renormalize the excited state energy such that the polaron shift is cancelled.

We are interested in the energetics, i.e., how the energy is distributed over time between the different terms in the total Hamiltonian. $\langle H_S(t)\rangle$ as well as $\langle H_{PS}(t)\rangle$ can be directly obtained from the reduced system density matrix. Because system observables are directly driven by the interaction term $H_I$, the observable closure for the environment observable $\hat{O}_1 = \sum_k \hbar g_k b_k^\dagger$ converges quickly ($\sum_k \hbar g_k b_k = \hat{O}_1^\dagger$ does not have to be calculated separately). Thus, we extract the mean interaction energy $\langle H_I(t)\rangle$ via the corresponding closure. Finally, the free phonon energy enters the equation of motion for the system observables only indirectly by affecting system-bath correlations. As discussed in the previous example, this implies that faster convergence is expected when the increase of the free phonon energy with respect to its initial value $\langle \Delta H_E^0(t)\rangle = \langle H_E^0(t)\rangle - \langle H_E^0(0)\rangle$ is not extracted directly but instead by integrating the equation of motion

$$\langle \Delta H_E^0(t)\rangle = -\frac{i}{\hbar} \int_0^t dt' \sum_k \langle [H, H_E^0]\rangle_{t'} = \int_0^t dt' \sum_k 2\hbar \omega_k g_k \mathrm{Im}\{\langle (b_k^\dagger |e\rangle\langle e|)\rangle_{t'}\} \,. \tag{67}$$

The right-hand side is obtained by constructing the environment observable closure for $\hat{O}_2 = \sum_k 2\hbar \omega_k g_k b_k^\dagger$.

First, we consider continuous resonant driving with $H_S = \frac{\hbar}{2}\Omega(|e\rangle\langle g| + |g\rangle\langle e|)$ with Rabi frequency $\Omega = 1$ ps$^{-1}$ turned on at time $t = 0$ for a two-level system initially in its ground state $\bar{\rho} = |g\rangle\langle g|$. For the phonon environment, we assume a spectral density $J(\omega) = \sum_k g_k^2\delta(\omega - \omega_k)$ of the form $J(\omega) = \omega^3\left(c_e e^{-\omega^2/\omega_e^2} - c_h e^{-\omega^2/\omega_h^2}\right)^2$ with $c_e = 0.1271$ ps$^{-1}$, $c_h = -0.0635$ ps$^{-1}$, $\omega_e = 2.555$ ps$^{-1}$, and $\omega_h = 2.938$ ps$^{-1}$, which is commonly used [14, 17, 18] to describe longitudinal acoustic phonons interacting with a semiconductor quantum dot in a GaAs matrix [63]. We discretize $J(\omega)$ using $N = 50$ modes equidistantly over the frequency range $[0, 7$ ps$^{-1}]$, and assume an initial bath temperature of $T = 4$ K. PT-MPOs are calculated using the ACE algorithm [23] for time steps $\Delta t = 0.1$ ps and compression threshold $\epsilon = 10^{-8}$.

The time evolution of the corresponding mean values of the energy terms in Eq. (66) are shown in Fig. 6(a). The mean system energy $\langle H_S\rangle$, which is due to the laser driving term, starts at zero and slowly becomes negative as the energy is redistributed to other terms. The polaron shift contribution $\langle H_{PS}\rangle$ is proportional to the excited state populations and thus shows slightly damped Rabi oscillations. The mean interaction energy is negative, which indicates binding between the excitation and the phonon cloud, i.e., polaron formation, and roughly mirrors the Rabi oscillations. The free phonon energy also oscillates, but also has an overall increasing trend, which indicates heating of the phonon bath. The change of the total energy with respect to its initial value $\langle\Delta H(t)\rangle = \langle H(t)\rangle - \langle H_E^0(0)\rangle$ remains constant because, after switching on the laser, the total Hamiltonian is constant in time and thus energy conserving. Only in the last few time steps are deviations found, which are again due to lack of convergence because of the small inner bonds at the ends of PT-MPOs. The conservation of the total energy serves as a crucial test for the physicality of the results and, thus, for the suitability of our approach to extract all relevant energetic contributions.

With this established, we now move to a more realistic but complex process, the phonon-assisted excitation of a quantum dot using a blue-detuned laser pulse. Such an excitation scheme enables bright and pure single-photon emission with frequency separation between excitation laser and emitted photons [14]. To this end, in Fig. 6(b) we present simulations using the total Hamiltonian Eq. (66) with system part

$$H_S = \frac{\hbar}{2}\big(\Omega(t)|e\rangle\langle g| + \Omega^*(t)|g\rangle\langle e|\big), \tag{68}$$

and pulse envelope $\Omega(t) = \frac{A}{2\pi\sigma}e^{-\frac{(t-t_0)^2}{2\sigma^2}}e^{-i\delta t}$,
and parameters $A = 3\pi$, $t_0 = 7$ ps, $\sigma = (5$ ps$)/\sqrt{8\ln 2}$, and $\delta = 1.5$ meV/$\hbar$. Furthermore, we use the same PT-MPOs as for Fig. 6(a).

Due to the external time-dependent driving, the total energy $\langle H\rangle$ is no longer conserved. From Fig. 6(b), it can be seen that most of the absorbed energy goes into the free phonon energy, i.e. in heating up the phonon bath. This is consistent with phonon-assisted excitation, where phonons are emitted during the photon absorption process in an effectively incoherent process, while there is only a slight build-up of system-environment correlations as measured by the interaction energy.

Finally, the dashed lines in Fig. 6(a) and (b) represent results, where, as discussed above, environment observable closures from ACE calculations are transferred to PT-MPOs obtained by the JP algorithm. The quantities obtained from the reduced system density matrix agree perfectly. The critical test, the mean interaction and free phonon energies, which involve extraction of information via inner bonds using observable closures also agree remarkably well, even if small quantitative deviations remain. This corroborates the thesis that the information conveyed via the inner bonds of PT-MPO, and hence the interpretation of the inner bonds themselves, is indeed universal and not only limited to PT-MPOs obtained from the ACE algorithm.

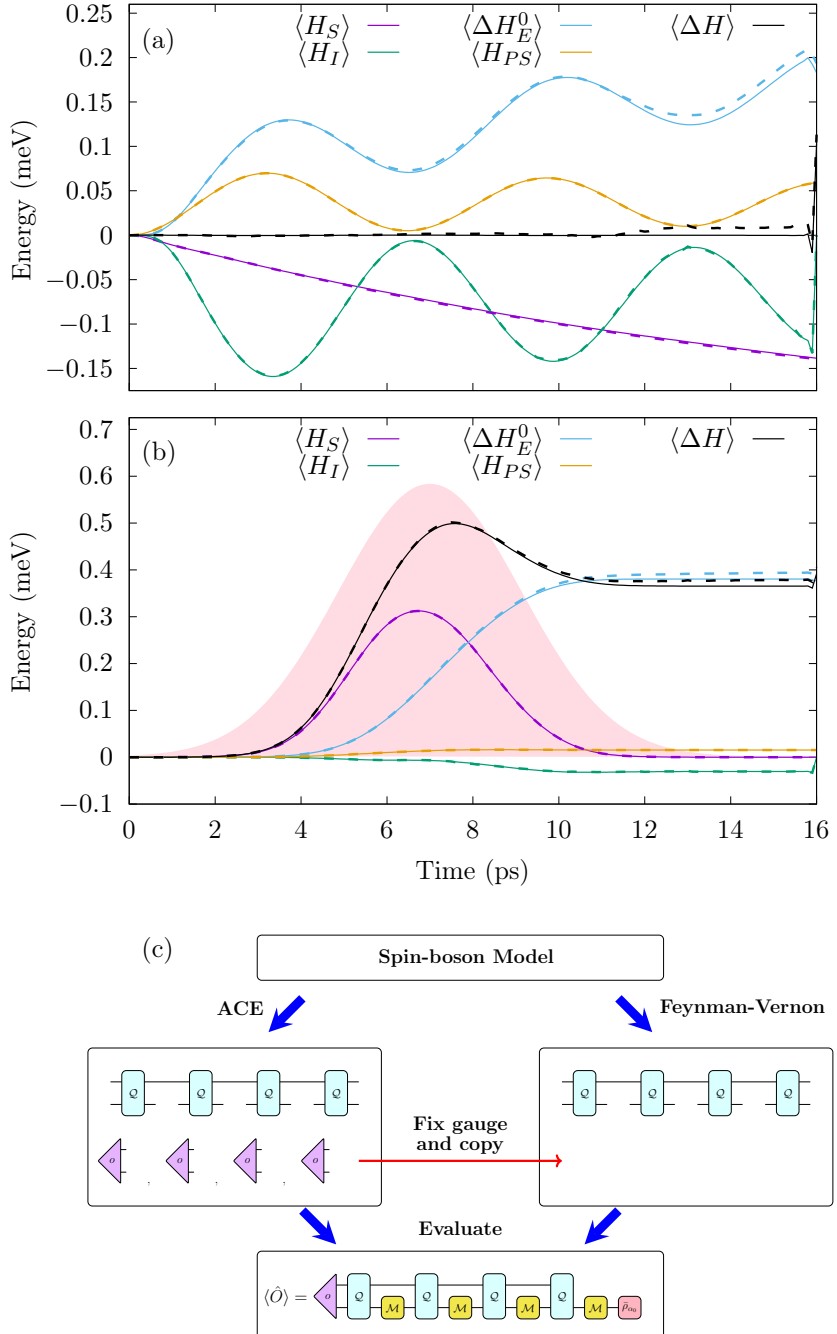

Figure 6: Time evolution of the means of the energy terms in Eq. (66) for a quantum dot coupled to phonons under (a) resonant continuous wave driving and (b) excitation by a blue-detuned Gaussian laser pulse ($|\hbar\Omega(t)|$ shaded in pink). The symbol $\Delta$ indicates that the initial value of the free phonon energy $\langle H_E^0(0)\rangle$ has been subtracted. Solid lines are obtained from ACE simulations using the observable closured described in the main text for $\epsilon = 10^{-8}$ and $\Delta t = 0.1$ ps. Dashed lines represent results of simulations, where the PT-MPOs are calculated using the algorithm by Jørgensen and Pollock (using Feynman-Vernon path integral expressions) in Ref. [25] and then transferring the observable closures obtained from the ACE simulations as sketched in (c).

# 5 Discussion

We have shown that the inner bonds of PT-MPOs are not merely a mathematical necessity required for describing the time-non-local memory in non-Markovian environments; they do in fact posses a concrete physical meaning: They directly represent the subspace of the full environment Liouville space containing the most relevant environment excitations, where the relevance is implicitly determined by MPO compression. Compression via truncated singular value decomposition minimizes the compression error without bias in favor of any particular set of system states. Hence, relevant environment degrees of freedom are determined without making any assumption about the concrete interventions on the system like (time-dependent) Hamiltonian evolution, Lindbladian losses, and projective measurement. It should be noted, however, that a more efficient representation of environmental influences may be available if interventions on the system are restricted. For example, transfer tensors [64] and small matrix decomposition of path integrals [65] provide extremely efficient numerically exact approaches when interventions are time-translation-invariant.

Conceptionally, the connection between the inner bonds of PT-MPOs and the environment Liouville space is expressed in terms of lossy linear transformation matrices $\mathcal{T}$ and their pseudoinverses $\mathcal{T}^{-1}$, both of which can be obtained by tracking all changes to the inner bonds at every step of the ACE algorithm [23]. With the help of these matrices, one can learn information about the state of the environment in PT-MPO simulations. However, it turns out to be more practical to extract environment observables $\hat{O}$ via a set of environment observable closures $\mathfrak{o}$, which transform like $\mathcal{T}^{-1}$ along the ACE algorithm, and which, when applied to PT-MPO inner bonds, represent the effect of $\mathrm{Tr}_E\{\hat{O}\rho(t)\}$. The viability and utility of this approach is tested on a series of scenarios involving the extraction of environment spins, currents, radiatively emitted photons, and energies absorbed by time-dependently driving an open quantum system strongly coupled to an environment.

However, because the compression is lossy, the extraction of an environment observable via inner bonds of PT-MPOs can be inaccurate if the subspace identified by MPO compression—designed to faithfully reproduce any system observable—does not carry the information required to reconstruct the environment observable. In such cases, alternative methods may be useful: Observables of certain environments are accessible via multi-time correlation function of the system [66]. More generally, environment observables can be obtained from methods that evolve the composite system and environment state, either by making physics-based approximations like in the reaction coordinate approach [67] or by numerically exact many-body representations as in chain mapping techniques [68–70].

On the other hand, the fact that different environment observables are extracted to different levels of accuracy enables us to probe what information is conveyed in the inner bonds of PT-MPOs. In particular, we find that observables which appear earlier in the hierarchy of Heisenberg equations of motions starting from expectation values of system observables converge faster with respect to the MPO compression parameter. This is in line with the fact that standard MPO compression selects environment degrees of freedom that most strongly affect the system evolution.

More broadly, our insights have significant consequences for fundamental and conceptual questions: First is the pedagogical aspect: Viewing PT-MPOs as the set of environment propagators $e^{\mathcal{L}_E \Delta t}$ projected onto the (locally) most relevant environment degrees of freedom via Eq. (1) is very intuitive. Yet this insight is sufficient for productively using existing PT-MPO-based open quantum systems codes [23,71], and practitioners need not understand Feynman-Vernon path integrals [19], the generalized Choi-Jamiołkowski isomorphism [39,40], or details about matrix product states in many-body quantum physics [21].

Second, Eq. (1) can facilitate a formal analysis and proofs. For example, it was straightforward to show in Eq. (58) that the composition of two PT-MPOs indeed provides a numerically exact way to simulate a quantum system coupled to two environments. A promising route for further progress is to analyze the transformations $\mathcal{T}$ and $\mathcal{T}^{-1}$ and the subspaces to which they map. This could lead to novel algorithms and it could clarify connections to other open quantum systems methods.

Finally, it is worth stressing that Eq. (1) constitutes a conceptional shift with respect to earlier derivations of PT-MPO methods [25]. The latter start from a time-non-local picture, where environment influences connecting system states at different points in time are represented efficiently. In contrast, in the picture developed here, the transformation matrix $\mathcal{T}^{-1}$ maps time-locally to the total system plus environment density matrix at a given point in time. Interestingly, we have also shown that environment observable closures calculated via ACE [23], which starts from the time-local picture, can be transferred to PT-MPOs obtained from the (time-non-local) bath correlation function [25], and the extracted environment observables are very similar. This suggests that the final PT-MPOs contain the same information, i.e. optimally compressed PT-MPOs are universal. So, the two pictures of time-non-local memory versus time-local environment excitations are intimately related, much like the Markov and the Born approximation tend to be used together for modelling weakly coupled open quantum systems [1]. While there is extensive literature on how to analyze and quantify non-Markovianity [47, 72–74], here, the focus on the compressed space of environment excitations suggests it is also worthwhile to consider measures of "non-Bornity". The universality of PT-MPOs further suggests that, for fixed time steps $\Delta t$, total number of time steps $n$, and MPO compression threshold $\epsilon$, the PT-MPO bond dimension $\chi$ is a promising candidate for such a measure of "non-Bornity".

## Acknowledgments

We would like to thank Jonathan Keeling, Brendon Lovett, and Gerald Fux for insightful discussions.

**Funding information** This work was supported from EPSRC Grants No. EP/T01377X/1 and No. EP/T007214/1. M.C. acknowledges funding by the Return Programme of the State of North Rhine-Westphalia.

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
