# Peer review of "Understanding and utilizing the inner bonds of process tensors"

_SciPost Physics, doi:SciPost Phys. 18, 024 (2025)_

## Round 1 · Referee Report · Anonymous (Referee 1) · 2024-5-27

Strengths

The manuscript introduces novel matrices \cal T and its pseudoinverse that facilitates insigths into the system and its environment. The formalism is illustrated by several physical examples.

Weaknesses

A possible weakness is the following. Section III B describes a compression of an MPO bond dimension. In the forward sweep subsequent tensors are svd-truncated. The backward sweep seems unnecesary as it only changes the gauge of the virtual indices. According to https://arxiv.org/pdf/1008.3477 sections 4.4.2 and 4.5.1, it would be optimal to use the forward sweep just to bring the MPO to the right-canonical form (without any truncations) and then to perform the truncations during the backwards sweep taking advantage of the mixed canonical form of the MPO. In this form each svd truncation is done in a tensor environment with a Euclidean metric tensor and the svd truncation is an optimal truncation. Would it be possible to implement this procedure in the manuscript?

Report

The manuscript is potentially acceptable. When the weakness is fixed the algorithm may become more powerful/stable.

Requested changes

There is one, see Weaknesses.

Recommendation

Ask for major revision

  • validity: high
  • significance: high
  • originality: good
  • clarity: good
  • formatting: good
  • grammar: excellent

Author:  Moritz Cygorek  on 2024-11-11  [id 4951]

(in reply to Report 1 on 2024-05-27)
Category:
answer to question

We are grateful to the Referee for their report and for raising this question. We agree that, as suggested in https://arxiv.org/pdf/1008.3477, for MPO compression with SVDs to be locally optimal, it would be necessary to first bring the MPO into the proper canonical form, which can be done, e.g., by sweeping with SVDs while not truncating in the forward sweep.

However, here we instead utilise a much faster, and whilst slightly suboptimal, yet entirely adequate strategy. This builds on the zip-up algorithm for MPS-MPO or MPO-MPO contractions by Stoudenmire and White [New J. Phys. 12, 055026 (2010)], which is also discussed in the review article mentioned by the Referee. Specifically, performing truncation already during forward sweeps leads to a significantly faster algorithm in practice. While it is not guaranteed to lead to optimal compression---in the sense that it produces minimal bond dimensions for a given accuracy---the results are nonetheless often found to be close to optimal. Additional errors incurred by this sub-optimality can be compensated by using a smaller compression threshold, yielding slightly larger bond dimensions. Overall, considering the trade-off between accuracy and computation time, the zip-up approach typically outperforms the nominally optimal compression, and has thus become the most common technique employed for PT-MPOs, such as in the first constructive PT-MPO algorithm for spin-boson models by Jorgensen and Pollock [Phys. Rev. Lett. 123, 240602 (2019)]. Another example is the precursor to PT-MPO methods, the TEMPO algorithm by Strathearn et al. [Nature Communications 9, 3322 (2018)].

The reason for the speed-up is that typical SVD routines require O(M^2N) operations, where M and N are the smaller and larger dimension of the matrix to be decomposed, respectively. When combining two MPOs with inner bond dimensions chi_1 and chi_2, for SVDs in the non-truncating sweep, one would have N=M=chi_1*chi_2. By contrast, the zip-up scheme involves SVDs on matrices where the small dimension M is already of the order of the inner bond dimension after compression chi. Thus, a relative speed-up by a factor of (chi_1*chi_2/chi)^2 is achieved.

It should be noted that the speed-up is so dramatic that, in most of our examples, convergence is out of reach if we do not use compression already in the forward sweep. However, it is possible to interpolate between nominally optimal compression and the zip-up approach, by using a smaller threshold for the forward sweep compared to the backward sweep. This is discussed in detail in Appendix A of Cygorek et al. [Phys. Rev. X 14, 011010 (2024)]. This also provides a practical solution strategy if suboptimal compression should ever result in issues for extracting environment observables.

Regarding the need for sweeps in the backward direction after the compression in the forward sweep, we point out that this is typically not only a gauge transformation. As also discussed in https://arxiv.org/pdf/1008.3477, even starting from the canonical form, truncating sweeps with SVDs are only locally optimal but truncation later in the chain is not independent of truncation earlier in the chain. Therefore, in many-body physics, one finds it necessary to sweep multiple times or to employ additional, e.g., variational techniques. Moreover, even the gauge transformation is useful, as it preconditions the MPO to a form much closer to the canonical form required for the next forward sweep. One way to see the importance of the backward sweep is that only when it is included, can one interpolate between nominally optimal compression and the zip-up scheme as described above.

We now discuss these subtle points in detail in Sec. III.B.

---

## Round 1 · Referee Report · Anonymous (Referee 2) · 2024-6-20

Report

The manuscript suggests a novel numerical method in open quantum systems. By introducing a lossy linear transformation, the authors build a link between the environment and the Process tensor matrix product operators. Most importantly, compare with traditional methods based on integrating out the environments, this linear transformation based on process tensor helps to resolve the environmental observable.

The authors tested carefully the validity of extracting environments from PT-MPO inner bonds in the sense that the compression is lossy. Demonstration of the method in spin system and the light matter coupled case is concrete. Specifically, there is a solid discussion on the stabilization of extracting photon number in the photon emission system. I strongly recommend publication of this manuscript.

Recommendation

Publish (easily meets expectations and criteria for this Journal; among top 50%)

  • validity: high
  • significance: good
  • originality: top
  • clarity: high
  • formatting: perfect
  • grammar: excellent

Author:  Moritz Cygorek  on 2024-11-11  [id 4952]

(in reply to Report 2 on 2024-06-20)
Category:
remark

We thank the Referee for their positive evaluation.

---

## Round 1 · Referee Report · Anonymous (Referee 3) · 2024-7-3

Strengths

1- Physical interpretation of performant algorithms for simulating open quantum systems. 2- Self-contained and clear explanations of the algorithms 3- Different numerical tests, showing strong evidence for their interpretation

Report

In this work, the authors elucidate the physical meaning of the "virtual" degrees of freedom in the MPO representation of the influence functional. This gives us insight into the success of different algorithms that are used for simulating open (non-Markovian) quantum systems. Moreover, their identification of these virtual degrees of freedom allows to evaluate environment expectation values approximately. The result is tested and confirmed on different benchmark cases. The paper is well-written and self-contained.

This work, therefore, provides a strong motivation for using these algorithms, and is a clear addition to the literature. I therefore recommend publication of this article.

Requested changes

1- For MPS, the singular value decomposition is optimal for truncating the virtual dimension in the MPS, because this optimizes the norm distance between the original and the truncated state. For MPOs, however, the choice for SVD truncation is not always straightforward, because it is not obvious which distance measure should be used. Could the authors argue why SVD truncation is a good choice here?

Recommendation

Publish (meets expectations and criteria for this Journal)

  • validity: high
  • significance: good
  • originality: good
  • clarity: high
  • formatting: perfect
  • grammar: perfect

Author:  Moritz Cygorek  on 2024-11-11  [id 4953]

(in reply to Report 3 on 2024-07-03)
Category:
answer to question

We thank the Referee for this interesting question. Indeed, we are not aware of any formal and useful (tight) bound for the error on any physical observable that is directly related to PT-MPO compression strategies and parameters.
Physical observables are obtained by eventually contracting the PT-MPO with a set of system propagators. PT-MPO are however designed to be independent of the particular choice of system propagators. One could in principle consider an adapted metric based on, e.g., a certain statistical distribution of system propagators. In this sense, the choice of considering the overall norm distance is unbiased and treats every possible system intervention with the same right.

Because of their novelty, error propagation in MPOs in the time domain are much less well understood than MPO representations in many-body physics. While analytical insights are lacking, one can turn to numerical experiments. For instance, in Appendix B of [https://arxiv.org/pdf/2405.16548] it was found that convergence of PT-MPOs has very similar trends whether one measures it via the PT-MPOs norm distance or one considers the changes in a concrete system observable. This suggests that minimising the norm distance is indeed a sensible criterion for PT-MPO compression.

We now provide more details on the rationale behind our choice of PT-MPO compression in Sec. III.B.

---

## Round 1 · Referee Report · Anonymous (Referee 4) · 2024-7-10

Report

The authors show how to extract physically relevant information from the inner bonds obtained from MPO compressions. This is a very interesting, relevant and timely contribution to the blossoming field of PT-MPO methods for open quantum system dynamics. It clearly highlights the meaning and relevance of the algorithms beyond their numerical value. Several examples (some somewhat very trivial, like the easily soluble amplitude damping model eq. (51)) are used to demonstrate the power of the PT-MPO method to reveal information about coupling and bath observables. A final remark: note that the two references [54,55] for the quantum thermodynamic application are not really concerned with coupled heat baths. There are far more suitable references for the scenario considered here.

Recommendation

Ask for minor revision

  • validity: -
  • significance: -
  • originality: -
  • clarity: -
  • formatting: -
  • grammar: -

Author:  Moritz Cygorek  on 2024-11-11  [id 4954]

(in reply to Report 4 on 2024-07-10)
Category:
answer to question

We thank the Referee for this remark. Indeed, the two references are concerned with how to define or measure thermodynamic quantities in the quantum setting without explicit reference to strong coupling. We have added a statement as well as a few references about the importance of methods that can treat strong system-environment couplings.

---

## Round 2 · Referee Report · Anonymous (Referee 1) · 2024-11-11

Report

I am grateful to the authors for a detailed answer to my comment. I do not want to hold this very interesting manuscript and recommend its publication in its present form.

However, I would like to point out that the MPO-MPS compression can be done in a locally optimal and efficient way by the zipper method described in Section IV of https://arxiv.org/pdf/2310.08533. One can have both the efficiency of the present manuscript and the mixed canonical form warranting the locally optimal truncation.

Recommendation

Publish (easily meets expectations and criteria for this Journal; among top 50%)

---

## Round 2 · Referee Report · Anonymous (Referee 4) · 2024-11-13

Report

I am happy with the changes of the revised version and recommend publication.

Recommendation

Publish (easily meets expectations and criteria for this Journal; among top 50%)

---

## Round 2 · Referee Report · Anonymous (Referee 3) · 2024-11-13

Report

The authors have responded adequately to the referee reports, so I support publication of this manuscript.

Recommendation

Publish (meets expectations and criteria for this Journal)

---

## Round 2 · Referee Report · Anonymous (Referee 2) · 2024-11-24

Report

I am satisfied with the revised version, and recommand publication.

Recommendation

Publish (easily meets expectations and criteria for this Journal; among top 50%)

---

## Round 2 · Author Response

We are happy to resubmit our manuscript "Understanding and utilizing the inner bonds of process tensors".
We are grateful to the Referees for their constructive and positive, and supportive reports. We have addressed the few remaining questions in our response and through revisions to the the manuscript.

We would like to acknowledge that we have identified an error in our treatment of the example where currents into fermionic environments are extracted. The treatment in the first version of our manuscript was correct for an environment consisting of two-levels treated as spins, but did not enforce the proper fermionic canonical anticommutation relations. Our resubmission has been delayed by the theoretical derivation, implementation, and numerical testing of an approach that fixes this issue, both for the calculation of fermionic PT-MPOs as well as for the extraction of currents. We have revised to corresponding section in the manuscript and updated the figure. Note, however, that the final results differ only in minor details, our conclusions remain entirely unaltered, and only a single example was affected.

---

## Round 2 · List of Changes

1) In Sec. III.B, we now provide a more detailed discussion and rationale for the particular MPO compression strategy used throughout our article.

2) We have revised the example on the extraction of currents in Sec. IV.B to now fully account for fermionic anticommutation relations.

3) We have added and updated references.

---

## Editorial Decision

published